# SPARK: Physics-Guided Quantitative Augmentation for Dynamical System Modeling

## Abstract

In dynamical system modeling, traditional numerical methods have a solid theoretical foundation but are limited by high computational costs and sensitivity to initial conditions. Current data-driven approaches use deep learning models to capture complex spatiotemporal features, but they rely heavily on large amounts of data and assume a stable data distribution, making them ineffective against data scarcity and distribution shifts. To address these challenges, we propose SPARK, a physics-guided quantized augmentation plugin. SPARK integrates boundary information and physical parameters, using a reconstruction autoencoder to build a physics-rich discrete memory bank for data compression. It then enhances selected samples for downstream tasks with this pre-trained memory bank. SPARK then utilizes an attention mechanism to model historical observations and combines fourier-enhanced graph ODE to efficiently predict long-term dynamical systems, enhancing robustness and adaptability to complex physical environments. Extensive experiments on benchmark datasets show that our approach significantly outperforms various baseline methods in handling distribution shifts and data scarcity.

## 1 Introduction

Modeling dynamical systems has long been a critical challenge across numerous scientific fields, including fluid dynamics (Li et al., 2023; Janny et al., 2023; Zhao et al., 2023), molecular dynamics (Brown & Yamada, 2013; Yang et al., 2022), and atmospheric science (Pathak et al., 2022; Bi et al., 2023), et al. Conventional numerical methods (Odibat & Baleanu, 2020), often rooted in rigorous partial differential equations (PDEs) (Long et al., 2018; Takamoto et al., 2022) and physical theory formula (Lippe et al., 2023), offer a robust foundation for modeling the dynamical evolution of complex systems. However, these methods are notoriously limited by their computational cost and sensitivity to different initial conditions and physical parameters.

Recently, numerous data-driven methods leveraging various neural network architectures have been proposed to solve this problem. They are committed to design delicate spatial and temporal components (Lippe et al., 2023; Raonic et al., 2024) to capture high-dimensional non-linear dynamical patterns and latent data distribution. This paper focuses exclusively on scenarios with fixed data observation points. Mainstream approaches select different model architectures based on whether the data is arranged in a regular pattern. Specifically, for irregular grids or complex geometric boundaries, graph neural networks (Kipf & Welling, 2017) are employed to capture intricate interactions between nodes and even along the boundaries (Wang et al., 2024a).

Despite their promising performance, the majority of these data-driven approaches considerably rely on a substantial volume of data and the assumption of distribution invariance (Wang et al., 2021; Yang et al., 2022). Formally, this dynamical system modeling task is still highly challenging in: (i) *Lack of Physical Guidance.* Some existing methods simply attach an external parameter embedding module (Lakshmikantham, 2019; Gao et al., 2021) to the neural networks. However, such methods struggle to capture higher-order correlations between physical parameters and data itself, and are difficult to generalize to unseen parameter configurations (Rame et al., 2022; Wu et al., 2024b). (ii) *Data Sparsity.* Data acquisition is limited due to the high computational cost of traditional numerical simulations (Schober et al., 2019) or the practical constraints on sensor usage in several real-world scenarios. (iii) *Out-of-distribution Generalization.*Within the dynamical systems, there usually exist two types of distribution shifts, namely environmental distribution shift (Li et al., 2022; Song et al.,

2023) and temporal distribution shift (Wang et al., 2022; Lu et al., 2024). The former is determined by predefined environmental attributes such as boundary conditions and physical parameters within the dynamical fields, while the latter arises from potential shifts within the data distribution over long-term temporal evolution.

To address these challenges, we propose a universal augmentation plugin, named SPARK, designed to efficiently encode rich physical priors alongside the given observations, enabling compressed representations and facilitating sample augmentation through physics-guided transformations. To begin, we extract latent representations of observed samples, where boundary information and physical parameters are fused through customized positional encodings and customized channel attention mechanisms. Further, to achieve efficient compression, we utilize a graph neural network framework combined with latent space quantization techniques (Van Den Oord et al., 2017), constructing a low-cost discretized memory bank infused with physical priors. Subsequently, we perform sample selection and query the pre-trained discretized memory bank for data augmentation, thus mitigating the impact of environmental distribution shift to some extent. Finally, to address temporal distribution shifts, we

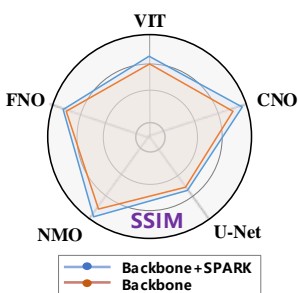

Figure 1: Performance comparison with or without SPARK on ERA5 dataset.

encode historical observations into an initial state through attention mechanism, and then implement a fourier-enhanced graph ODE for effectively long-term prediction. As shown in Figure 1, augmenting data through our SPARK can consistently improve the performance of baseline models.

In summary, our paper makes the following contributions: (1) *Novel Perspective.* We are the first to propose a physics-guided compression and augmentation plugin, which highly enhances the generalization capability to diverse physical scenarios. (2) *Modern Architecture.* In downstream task, we encode historical observations into a latent space based on attention mechanism and introduce a fourier-enhanced graph ODE to overcome temporal distribution shifts and realize efficient long-term predictions. (3) *Verification.* We demonstrate the generalization and robustness of SPARK under data scarcity and distribution shifts through both experimental evaluations and theoretical analysis.

## 2 RELATED WORK

**Dynamical System Modeling.** Deep neural networks have recently emerged as powerful tools for tackling the challenges associated with dynamics forecasting (Gao et al., 2022; Yin et al., 2022), showcasing their capabilities to efficiently model intricate, high-dimensional systems. To handle complex spatiotemporal dependencies, various advanced models based on convolutional neural networks (CNN) (Ren et al., 2022; Raonic et al., 2024), recurrent neural networks (RNN) (Mohajerin & Waslander, 2019; Maulik et al., 2021), Transformer (Wu et al., 2023a; Chen et al., 2024), or exquisite hybrid architectures (Shi et al., 2015; Wu et al., 2023b) have been proposed. Recently, Neural Operators (Li et al., 2021; Rahman et al., 2022; Tran et al., 2023) become a popular data-driven approach by learning to approximate the past-future infinite-dimensional function space mappings. Additionally, Physics-Informed Neural Networks (PINN) (Raissi et al., 2019; Wang et al., 2020) integrate prior physical knowledge as additional regularizers into the training process for physical constraints, particularly in systems governed by partial differential equations. Moreover, most of these approaches are limited to structured grids and lack the ability to handle irregular grids or varying connectivity. Thus researchers have turned to graph neural networks (Fan et al., 2019) as a promising alternative to accommodate a broader range of scenarios. GNNs inherit key physical properties from geometric deep learning, including permutation invariance and spatial equivariance (Li et al., 2020; Wu et al., 2022), which offer distinct advantages for modeling dynamical systems.

**Out-of-distribution Generalization.** Out-of-distribution (OOD) generalization (Liu et al., 2021a; Hendrycks et al., 2021; Deng et al., 2023) has emerged as a critical challenge in machine learning, especially for models that encounter distribution shifts between training and testing data. Significant advancements have been made in OOD generalization techniques, containing invariant causal inference (Gui et al., 2023; Luo et al., 2024), data augmentation (Wang et al., 2024b), and domain adaptation (Kundu et al., 2020; Garg et al., 2022), et al. Most existing work focuses on static scenarios (Li et al., 2022; Gui et al., 2023), with limited research on dynamical systems. These dynamic

Figure 2: **Model overview** – The proposed framework consists of four steps: (1) Physical prior incorporation, where boundary and physical parameters are encoded into the input observations; (2) Dynamical discretization modeling through reconstruction to create a discrete physics-rich memory bank; (3) Memory bank promoted augmenting on sampled training data; and (4) Fourier-enhanced graph ODE for dynamical system prediction based on historical observations.

systems exhibit more complicated distribution patterns corresponding to varying system properties or spatiotemporal environments, which is the primary focus of our study. In this work, we propose SPARK, which compresses rich physical information into a discrete memory bank for generalization, and design a fourier-enhanced graph ODE to relieve temporal distribution shifts.

## 3 METHODOLOGY

**Problem Definition.** Given a dynamical system governed by physical laws such as PDEs, we aim to enhance prediction using autoencoder reconstruction and discrete quantization. We have $N$ observation points in the domain $\Omega$, located at $\mathbf{s} = \{\mathbf{s}_1, \cdots, \mathbf{s}_N\}$, where $\mathbf{s}_i \in \mathbb{R}^{d_s}$. At time step $t$, the observations are $\mathcal{X}^t = \{\mathcal{X}_1^t, \cdots, \mathcal{X}_N^t\}$, where $\mathcal{X}_i^t \in \mathbb{R}^d$ and $d$ represents the number of observation channels. Boundary information and physical parameters affect the dynamical system, leading to different conditions and distribution shifts. We first employ reconstruction model and construct a discrete memory bank to compress and store physical prior information. Then, given historical observation sequences $\{\mathcal{X}_i^{-T_0+1:0}\}_{i=1}^N$, our goal is to use the pre-trained memory bank for data augmentation and predict future observations $\{\mathcal{Y}_i^{1:T}\}_{i=1}^N$ at each observation point.

### 3.1 FRAMEWORK OVERVIEW

In this section, we systematically introduce our SPARK framework, as shown in Figure 2. We first introduce a coupled reconstruction autoencoder, which incorporates boundary information and physical parameters, to generate a discrete, physics-rich memory bank for compression. Subsequently, in downstream task, we employ the pre-trained memory bank to augment the training samples based on the fusion mechanism. We then utilize the attention mechanism to model historical observations and propose a fourier-based graph ODE to realize downstream dynamical system prediction.

### 3.2 PHYSICS-INCORPORATED DATA COMPRESSION

Here, we seek to transform the dynamical system observations, along with rich boundary information and physical parameters, into a structured representation. We then maintain a discrete memory bank and leverage graph neural networks to reconstruct the observations based on this representation.

**Boundary Information Incorporation.** Before merging the boundary information into the historical observations, it is important to construct the graph $\mathcal{G} = \{\mathcal{V}, \mathcal{E}\}$ with the finite discrete internal node. Similar to other methods within spatial domian (Fan et al., 2019), we propose to select the $K$-nearest nodes for each node to construct the edge set $\mathcal{E}$.

Then, how to effectively embed boundary information into neural networks is a critical aspect of the design. From prior research (Wang et al., 2024a; Lötzsch et al., 2022), the straightforward concatenation of boundary information with node features has been regarded as suboptimal and is easily overfit. Here, we integrate the relative positional information with the boundary into each node's positional encoding, which is further projected into the node features.

$$\boldsymbol{u}_i = \text{Proj}\left(\mathcal{X}_i, \boldsymbol{p}_i^{rel}\right) \quad \text{with} \quad \boldsymbol{p}_i^{rel} = \phi\left(\mathbf{s}_i, \boldsymbol{p}_i^{boun}\right), \tag{1}$$

where $\mathbf{s}_i$ is the location information of node $i$, and $\boldsymbol{p}_i^{boun}$ denotes the relative positional relationship between node $i$ and the nearest boundary point. For better incorporation, we embed the complete boundary information into a latent vector $\mathcal{B}$ using an MLP, which are then fed into each message-passing layer of the GNN during the subsequent reconstruction phase.

**Physical Parameters Guided Channel Attention.** Inspired by (Takamoto et al., 2023), we employ a channel attention mechanism to effectively embed external parameters into the neural networks, facilitating the transfer of parameter information into the latent space. Hereafter, we refer to the neural network parameters as 'weights' to avoid terminological conflicts with the physical parameters in dynamical systems. Specifically, the channel attention obtains two $d$-dimensional mask attention vectors $\boldsymbol{a}_\vartheta \in \mathbb{R}^d$ ($\vartheta = 1, 2$) from the parameters $\boldsymbol{\delta}$ using a 2-layer MLP.

$$\boldsymbol{a}_\vartheta = \boldsymbol{W}_{2,\vartheta}\sigma(\boldsymbol{W}_{1,\vartheta}\boldsymbol{\delta} + \boldsymbol{b}_1) + \boldsymbol{b}_2, \tag{2}$$

where $\boldsymbol{W}_{1/2,\vartheta}$ is the weight matrix, $\boldsymbol{b}_{1/2}$ is the bias term, and $\sigma$ is the GeLU activation function. For the boundary-enhanced node feature $\boldsymbol{u}$, we employ two convolutional opertors: a $1 \times 1$ convolution ($g_1$) and a spectral convolution ($g_2$). The former one ensures fine-grained alignment between channels, and the latter one operates at a global frequency-domain and captures the broader and structural correlations. Then the obtained representations are multiplied by the mask attention vectors separately and finally combined to realize the channel attention between observations and parameters.

$$\boldsymbol{z}_{i,\vartheta} = g_\vartheta(\boldsymbol{u}_i), \quad \boldsymbol{h}_{i,\vartheta} = \boldsymbol{a}_\vartheta \odot \boldsymbol{z}_{i,\vartheta} \quad \Rightarrow \quad \boldsymbol{h}_i = \boldsymbol{u}_i + \boldsymbol{h}_{i,1} + \boldsymbol{h}_{i,2}, \tag{3}$$

where $\odot$ is the Hadamard operator. By this way, we achieve parameter channel fusion for physical processes through attention-enhanced convolutional networks.

**Physics-Embedded Reconstruction with Discrete Memory bank.** Now we have obtained the observation features $\boldsymbol{h}$ fusing with the boundary and parameter information. For reconstruction, we propose to tokenize each node as discrete embeddings using a $L$-layer GNN encoder and a variant of VQ-VAE (Van Den Oord et al., 2017). Formally, in the $l$-th layer of the GNN encoder, the output representations $\mathbf{h}^{(l)}$ are combined from the previous representations using aggregation operator and residual function as:

$$\boldsymbol{h}_i^{'(l)} = \text{AGGREGATE}^{(l)}\left(\left\{\boldsymbol{h}_j^{(l-1)} : j \in \mathcal{N}(i)\right\}, \mathcal{B}\right), \boldsymbol{h}_i^{(l)} = \text{COMBINE}^{(l)}\left(\boldsymbol{h}_i^{(l-1)}, \boldsymbol{h}_i^{'(l)}\right), \tag{4}$$

where $\mathcal{B}$ denotes the aforementioned boundary latent vector. Then, the selector look up the nearest neighbor code embedding $\boldsymbol{z}_i$ in the maintained memory bank $\boldsymbol{E} = [\boldsymbol{e}_1, \boldsymbol{e}_2, \cdots, \boldsymbol{e}_M] \in \mathbb{R}^{M \times D}$ ($M$ denotes the memory bank size) for each node embedding $\mathbf{h}_i$.

$$\boldsymbol{z}_i = \text{argmin}_j \left\|\boldsymbol{h}_i^L - \boldsymbol{e}_j\right\|_2. \tag{5}$$

Finally, the discretized representations $\boldsymbol{z}$ are fed into a linear decoder $\mathcal{I}(\cdot)$ to reconstruct the input observations for an end-to-end optimization.

**Pre-training Loss Function.** For the whole reconstruction pre-training, we minimize the reconstruction loss and the discrete memory bank loss simultaneously. Specifically, to overcome the training challenges associated with discretization encoding, we utilize the stopgradient operator $\mathbf{sg}(\cdot)$ (Van Den Oord et al., 2017). The whole pre-training loss function $\mathcal{L}_{pre}$ is as follows:

$$\mathcal{L}_{pre} = \frac{1}{TN}\sum_{t=1}^{T}\sum_{i=1}^{N}\left(\hat{\mathcal{X}}_i^t - \mathcal{X}_i^t\right)^2 + \frac{1}{TN}\sum_{t=1}^{T}\sum_{i=1}^{N}\left(\mu\left\|\boldsymbol{h}_i^t - \mathbf{sg}[\boldsymbol{e}]\right\|_2^2 + \gamma\left\|\mathbf{sg}\left[\boldsymbol{h}_i^t\right] - \boldsymbol{e}\right\|_2^2\right), \tag{6}$$

where $\hat{\mathcal{X}}_i^t$ and $\mathcal{X}_i^t$ denote the reconstructed and the initial node embedding, $\mu$ and $\gamma$ are the hyperparameters to balance the loss components.

### 3.3 DOWNSTREAM DATA AUGMENTATION AND DYNAMICS SYSTEM PREDICTION TRAINING

**Memory bank Promoted Augmentation.** Now we have compressed multi-physics coupled information into a discrete memory bank. Based on this, we introduce a memory bank promoted data augmentation strategy to enhance the robustness and generalization of the training set. Specifically, given a training set $\mathcal{D} = \{(\mathcal{X}_i, \mathcal{Y}_i)\}_{i=1}^N$, where $\mathcal{X}_i$ represents the input dynamical system observations and $\mathcal{Y}_i$ denotes the corresponding output (e.g., future states), we augment the dataset by sampling a subset $\mathcal{D}' \subset \mathcal{D}$ according to a probabilistic sampling method. Formally, for each sampled data, we utilze the pre-trained GNN encoder and follow the equation (5) to augment the data by combining with its Top-K nearest discrete embeddings:

$$\boldsymbol{v}_i = \lambda \boldsymbol{h}_i + (1 - \lambda) \sum_{n=1}^{K} \boldsymbol{e}_n, \tag{7}$$

where $\lambda \in [0, 1]$ is the balance factor. Using Top-K nearest neighbors for augmentation enhances physical diversity and better captures the underlying data distribution. The augmented samples are incorporated into the training set to enhance the model's generalization performance during subsequent training stage.

**Fourier-enhanced Graph ODE for Prediction.** We first manage to map the historical observations of individual node into a latent representation. Specifically, we employ an attention mechanism to calculate the corresponding attention score of each time step, and then initialize the observation state vector based on this. In formulation:

$$\boldsymbol{q}_i = \frac{1}{T_0} \sum_{t=1}^{T_0} \delta(\alpha_i^t \cdot \boldsymbol{v}_i^t), \quad \alpha_i^t = \left(\boldsymbol{v}_i^t\right)^T \cdot \tanh\left(\left(\frac{1}{T_0} \sum_{t=1}^{T_0} \boldsymbol{v}_i^t\right) W_\alpha\right), \tag{8}$$

where $\alpha_i^t$ denotes the attention score of node $i$ at time step $t$. We further propose a fourier-enhanced graph ODE to model the obtained state vector by integration of the spectral global feature and the local feature. In formulation:

$$\frac{d\boldsymbol{q}_i}{dt} = \Phi\left(\boldsymbol{q}_1, \boldsymbol{q}_2 \cdots \boldsymbol{q}_N, \mathcal{G}, \Theta\right) = \sum_{l=1}^{L} \sigma\left(\mathcal{F}^{-1}\left(A\mathcal{F}\left(H^l\right)W_\mathcal{F}^l\right) + AH^lW^l + b^l\right). \tag{9}$$

Here, $\mathcal{G}$ and $\Theta$ are the graph structure and the whole model parameters, $\mathcal{F}$ denotes *Discrete Fourier Transform (DFT)* (Tolstov, 2012), $A$ is the adjacency matrix, and $W_\mathcal{F}$, $W$, $b$ are the matrix weights. Then, the latent dynamics can be solved by any ODE solver like Runge-Kutta (Schober et al., 2014):

$$\boldsymbol{q}_i^{t_1} \cdots \boldsymbol{q}_i^{t_T} = \text{ODESolve}(\Phi, [\boldsymbol{q}_1^0, \boldsymbol{q}_2^0 \cdots \boldsymbol{q}_N^0]) \implies \widehat{\boldsymbol{y}}_i^t | \boldsymbol{q}_i^t \sim p(\widehat{\boldsymbol{y}}_i^t | f_{\text{dec}}(\boldsymbol{q}_i^t)), \tag{10}$$

where $\{\boldsymbol{q}_i^{t_1} \cdots \boldsymbol{q}_i^{t_T}\}$ represents the latent future vector of node $i$, and $\widehat{\boldsymbol{y}}_i^t$ is the corresponding predicted vector through a stacked two-layer MLP decoder $f_{dec}(\cdot)$.

**Training.** The training objective for the dynamics system prediction is to learn a mapping from the input observations $\mathcal{X}$ to future states $\mathcal{Y}$, where the system evolves according to certain physical laws like parameters. Formally, we aim to minimize the following loss function:

$$\mathcal{L}_{\text{dyn}} = \frac{1}{TN} \sum_{i=1}^{T} \sum_{i=1}^{N} \|\hat{\mathcal{Y}}_i^t - \mathcal{Y}_i^t\|_2^2 + \lambda_{\text{reg}} \mathcal{R}(\theta), \tag{11}$$

where $\hat{\mathcal{Y}}$ denotes the predicted future states, and $\mathcal{R}(\theta)$ is a regularization term on the model parameters $\theta$ to prevent overfitting. Finally, to seamlessly integrate the augmented samples into the training pipeline, we adopt a curriculum learning approach, gradually adding the augmented samples into the training set. Initially, the model is trained on the original dataset, and as training progresses, the proportion of augmented samples is increased, allowing the model to adapt to a richer variety of dynamical behaviors.

### 3.4 THEORETICAL ANALYSIS

**Theorem 1** (*Enhancement of Model Generalization via Physical Priors from an Information-Theoretic Perspective*). *Let $\mathcal{D}$ be the training dataset, $\theta$ be the model parameters, and $\mathcal{P}$ be the physical*

*prior information. Assume that the conditional mutual information between $\theta$ and $\mathcal{D}$ given $\mathcal{P}$ is $I(\theta; \mathcal{D} \mid \mathcal{P})$.*

*For an i.i.d. training dataset of size $n$, the upper bound on the expected generalization error is:*

$$\left| \mathbb{E}_{\theta, \mathcal{D}} \left[ \mathcal{L}(\theta) - \mathcal{L}_{emp}(\theta; \mathcal{D}) \right] \right| \leq \sqrt{\frac{2I(\theta; \mathcal{D} \mid \mathcal{P})}{n}}, \tag{12}$$

*where $\mathcal{L}(\theta)$ is the expected loss under the true distribution, and $\mathcal{L}_{emp}(\theta; \mathcal{D})$ is the empirical loss on the training data.*

Thus, introducing physical prior information $\mathcal{P}$ reduces the conditional mutual information $I(\theta; \mathcal{D} \mid \mathcal{P})$, which, by the above theorem, decreases the upper bound of the generalization error. This implies that physical priors enhance model generalization, improving performance on unseen data. The detailed theorem with the proof is illustrated in Appendix A.

**Theorem 2** (***Upper Bound on Generalization Error in Bayesian Learning with Physical Priors***). *Let $\mathcal{H}$ be a hypothesis space, $\theta \in \mathcal{H}$ be the model parameters, and $\mathcal{D} = \{(\mathcal{X}_i, \mathcal{Y}_i)\}_{i=1}^{N}$ be the training dataset. Let $\ell(\theta; \mathcal{X}, \mathcal{Y})$ be the loss function, with the true risk defined as $\mathcal{L}(\theta) = \mathbb{E}_{(\mathcal{X}, \mathcal{Y}) \sim \mathcal{P}_{data}}[\ell(\theta; \mathcal{X}, \mathcal{Y})]$ and the empirical risk as $\mathcal{L}_{emp}(\theta) = \frac{1}{N} \sum_{i=1}^{N} \ell(\theta; \mathcal{X}_i, \mathcal{Y}_i)$.*

*Assume the prior distribution $P(\theta)$ incorporates physical prior information, and the posterior distribution is $Q(\theta)$. For any $\delta > 0$, with probability at least $1 - \delta$, the following upper bound on the generalization error holds:*

$$\mathbb{E}_{\theta \sim Q}[\mathcal{L}(\theta)] \leq \mathbb{E}_{\theta \sim Q}[\mathcal{L}_{emp}(\theta)] + \sqrt{\frac{KL(Q\|P) + \ln \frac{2\sqrt{N}}{\delta}}{2N}}, \tag{13}$$

*where $KL(Q\|P)$ is the Kullback-Leibler divergence (Van Erven & Harremos, 2014) between the posterior distribution $Q$ and the prior distribution $P$.*

By incorporating physical prior information as the prior $P(\theta)$, the KL divergence $KL(Q\|P)$ between the posterior $Q(\theta)$ and the prior $P(\theta)$ is reduced, thereby lowering the upper bound on the generalization error. This demonstrates that physical priors enhance model generalization in the Bayesian framework. We have the theorem with proof in Appendix B.

## 4 EXPERIMENT

In this section, we present experimental results to demonstrate the effectiveness of the SPARK framework. Our experiments are designed to address the following research questions: **RQ1.** Does SPARK effectively handle out-of-distribution generalization while maintaining consistent superiority? **RQ2.** Can SPARK tackle challenging tasks? **RQ3.** Is SPARK scalable and physically consistent? **RQ4.** Does SPARK enhance model generalization?

### 4.1 EXPERIMENTAL SETTINGS

**Benchmarks.** We choose benchmark datasets from three fields. First, for computational fluid dynamics, we use **Prometheus** (Wu et al., 2024b) and follow its original environment settings. Second, for real-world data, we use **ERA5** (Hersbach et al., 2020), selecting different combinations of atmospheric $u, v$ velocity components and $humidity$ as forcing terms to predict atmospheric temperature. Finally, for partial differential equations, we examine the **2D Navier-Stokes Equations** (Li et al., 2021), focusing on the effect of viscosity $\nu$ on vorticity, and simulate vorticity under ten different viscosities. We also study the **Spherical Shallow Water Equations** (Galewsky et al., 2004) to simulate large-scale atmospheric and oceanic flows on Earth's surface, involving viscosity $\nu$, tangential vorticity $w$, and fluid thickness $h$. Additionally, we consider the **3D Reaction-Diffusion Equations** (Rao et al., 2023), describing chemical diffusion and reaction in space, with diffusion coefficient $D$, and involving $u$ and $v$ velocity components. More detailed descriptions see Appentix D.

**Baselines.** We select representative models from two domains as baselines. ▷ **Visual Backbone Networks.** We include ResNet (He et al., 2016), U-Net (Ronneberger et al., 2015), Vision Transformer

Table 1: Comparison of different models on five benchmark datasets (Prometheus, Navier–Stokes, Spherical-SWE, 3D Reaction–Diff, ERA5) with and without OOD. Our method (Ours + SPARK) achieves the best performance across all benchmarks, especially under OOD conditions.

| | BENCHMARKS | | | | | | | | | |
| MODEL | PROMETHEUS | | NAVIER–STOKES | | SPHERICAL-SWE | | 3D REACTION–DIFF | | ERA5 | |
| | *w/o* OOD | *w/* OOD | *w/o* OOD | *w/* OOD | *w/o* OOD | *w/* OOD | *w/o* OOD | *w/* OOD | *w/o* OOD | *w/* OOD |
|---|---|---|---|---|---|---|---|---|---|---|
| U-NET 2015 | 0.0931 | 0.1067 | 0.1982 | 0.2243 | 0.0083 | 0.0087 | 0.0148 | 0.0183 | 0.0843 | 0.0932 |
| RESNET 2016 | 0.0674 | 0.0696 | 0.1823 | 0.2301 | 0.0081 | 0.0192 | 0.0151 | 0.0186 | 0.0921 | 0.0977 |
| VIT 2021 | 0.0632 | 0.0691 | 0.2342 | 0.2621 | 0.0065 | 0.0072 | 0.0157 | 0.0192 | 0.0762 | 0.0786 |
| SWINT 2021B | 0.0652 | 0.0729 | 0.2248 | 0.2554 | 0.0062 | 0.0068 | 0.0155 | 0.0190 | 0.0782 | 0.0832 |
| FNO 2021 | 0.0447 | 0.0506 | 0.1556 | 0.1712 | 0.0038 | 0.0045 | 0.0132 | 0.0179 | 0.7233 | 0.9821 |
| UNO 2022 | 0.0532 | 0.0643 | 0.1764 | 0.1984 | 0.0034 | 0.0041 | 0.0121 | 0.0164 | 0.6652 | 0.7621 |
| CNO 2024 | 0.0542 | 0.0655 | 0.1473 | 0.1522 | 0.0037 | 0.0038 | 0.0145 | 0.0182 | 0.5243 | 0.7821 |
| NMO 2024C | 0.0397 | 0.0483 | 0.1021 | 0.1032 | 0.0026 | 0.0031 | 0.0129 | 0.0168 | 0.0432 | 0.0563 |
| OURS + SPARK | **0.0294** | **0.0308** | **0.0714** | **0.0772** | **0.0018** | **0.0020** | **0.0102** | **0.0116** | **0.0322** | **0.0321** |

Figure 3: **Comparison of Prediction Performance of Different Models Over Time Evolution.** The figure shows the target values and the predictions from different models (SPARK, FNO, VIT) at multiple time steps. It is evident that SPARK's predictions are closest to the target values, especially in the locally complex regions (highlighted in red or white boxes), demonstrating higher detail-capturing ability and accuracy. In contrast, FNO and VIT show larger deviations in the same regions.

(ViT) (Dosovitskiy et al., 2021), and Swin Transformer (SwinT) (Liu et al., 2021b). ▷ **Neural Operator Architectures.** We consider FNO (Li et al., 2021), UNO (Ashiqur Rahman et al., 2022), CNO (Raonic et al., 2024), and NMO (Wu et al., 2024c). More details see Appentix E.

## 4.2 ASSESSING THE EFFICACY OF SPARK (RQ1)

As shown in Table 1 and Figure 3, We have the following observations:

**Obs 1:** Our method (SPARK) shows better performance on all benchmark datasets, especially with out-of-distribution (OOD) data. For example, in the Prometheus dataset, SPARK achieves an MSE of 0.0294 and 0.0308 for regular and OOD conditions, respectively, significantly outperforming the next best model, NMO (0.0483 and 0.0483), with an error reduction of about 36%. On the Navier-Stokes dataset, SPARK achieves an MSE of 0.0714 (regular) and 0.0772 (OOD), lower than NMO's 0.1021 and 0.1032. On the Spherical-SWE dataset, SPARK achieves an MSE of 0.0018 and 0.0020, also outperforming other models like FNO (0.0038 and 0.0045). For the ERA5 dataset, SPARK's MSE (0.0322 and 0.0321) is better than models like ResNet and FNO, demonstrating SPARK's superior adaptability and robustness across different conditions.

**Obs 2:** SPARK shows better performance under both regular and out-of-distribution (OOD) conditions on all benchmark datasets, clearly outperforming existing baseline models. This improvement is most evident in handling complicated fluid dynamics data (e.g., Navier-Stokes and Spherical-SWE) and real meteorological data (e.g., ERA5), where SPARK's error is significantly lower than other models, proving its strong generalization ability and adaptability.

**Obs 3:** The visualization results, as shown in the figure 3, demonstrate that in the ERA5 and Navier-Stokes experiments, SPARK outperforms other models (such as FNO (Li et al., 2021) and VIT (Dosovitskiy et al., 2021)) in capturing complex details of fluid dynamics. Particularly in specific

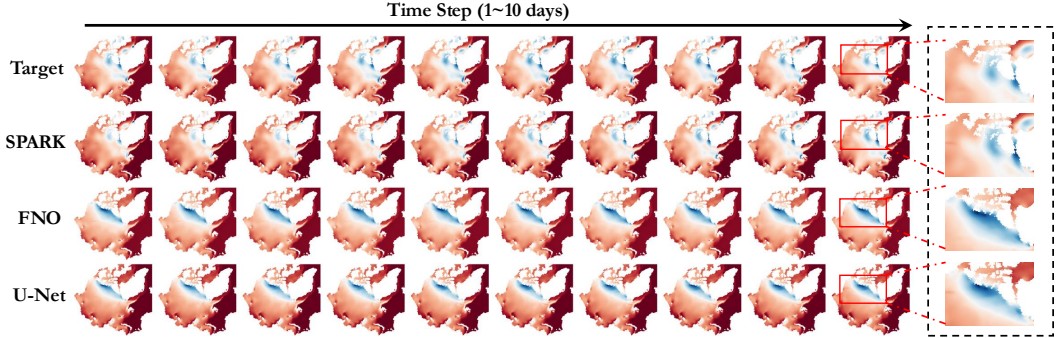

Figure 4: The figure shows the predictions of different models for sea ice data over 1 to 10 time steps. The horizontal axis represents time steps, and the vertical axis shows, from top to bottom, the target data, SPARK, FNO, and U-Net model predictions. SPARK predictions are closer to the target, while FNO and U-Net have larger deviations.

regions, Spark's predictions are closer to the target and capture details more accurately. This indicates that Spark has stronger generalization and robustness in complex spatiotemporal dynamic predictions.

These results show that SPARK, with its physics-guided data compression and augmentation mechanism, effectively improves the model's generalization and robustness under scarcity data and distribution shift conditions.

### 4.3 SPARK CAN HANDLE CHALLENGING TASKS EFFECTIVELY (RQ2)

Here, we analyze SPARK's performance on **sea ice** data based on experimental results, as shown in Figure 4 and Figure 5. The **sea ice** data is derived from ERA5. This challenge arises from the complex, nonlinear interactions governing its Lagrangian motion (Notz, 2012), compounded by the spatiotemporal variability of environmental forcing factors.

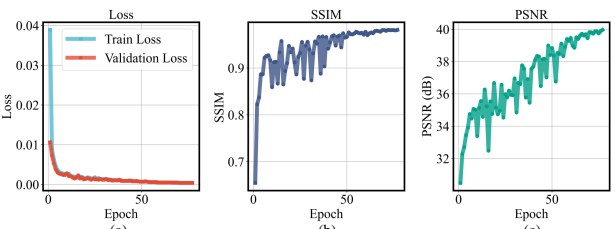

Figure 5: Performance Metrics for SPARK.

**Obs 1: Qualitative Analysis.** In the visual results of sea ice prediction (Figure 4), SPARK's predictions are closest to the target values, especially in locally complex regions, demonstrating strong detail-capturing ability and prediction accuracy. In contrast, FNO and U-Net show larger deviations in the same regions, particularly at boundaries and areas with significant structural changes, indicating SPARK's better generalization and robustness in handling complex dynamic systems.

**Obs 2: Quantitative Analysis.** From the loss curve during training (Figure 5 a), SPARK's training and validation losses rapidly decrease and stabilize within 80 epochs, indicating a good fit on both training and validation sets. Additionally, SSIM and PSNR (Figures 5 b and 5 c) increase with epochs, eventually approaching high values of approximately 0.95 and 40 dB, respectively. This means the model performs well in reconstruction quality and image clarity, capturing complex spatiotemporal features and providing high-fidelity predictions.

Both qualitative and quantitative analyses demonstrate that SPARK not only surpasses other models (like FNO and U-Net) in numerical evaluation metrics but also more precisely captures dynamic changes in complex scenarios, such as long-term sea ice evolution. This highlights its effectiveness and distinct advantage in tackling challenging tasks.

### 4.4 ANALYSIS OF PHYSICAL CONSISTENCY AND SCALABILITY (RQ3)

In this section, we focus on examining the physical consistency and scalability of the SPARK framework. Drawing from the energy spectrum visualization in Figure 6 and the performance metrics of pretrained models of different sizes in Table 2, we highlight two key observations:

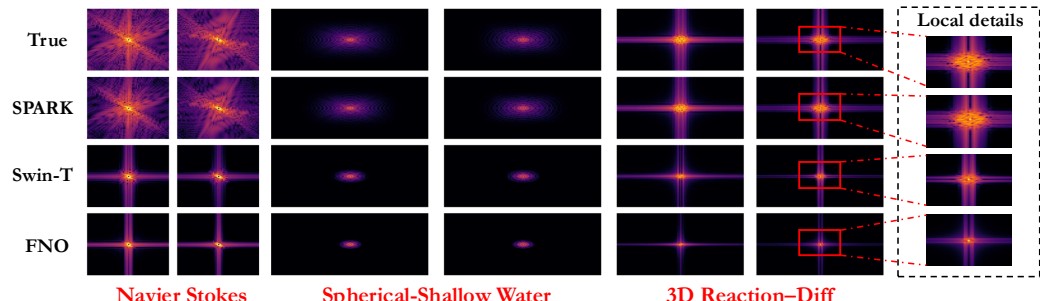

Figure 6: Energy Spectrum Comparison Results.

**Obs 1: SPARK shows excellent physical consistency, with energy spectra closer to real data.**
Figure 6 shows that the energy spectra generated by SPARK are very close to real data, especially in complex fluid dynamics scenarios like Navier-Stokes and Spherical Shallow Water. Compared to other baseline models such as Swin-T and FNO, SPARK better captures the complex details of dynamic systems. This performance is due to SPARK incorporating physical priors, which effectively enhances its ability to model physical phenomena and improves its physical consistency.

**Obs 2: Model performance remains stable as size reduces, showing good scalability.** The Table 2 shows that pretrained models of different sizes have similar error rates on the ERA5 and Navier-Stokes datasets. For example, when the model size decreases from 24.56 MB to 9.43 MB, the error on ERA5 only increases slightly from 0.0302 to 0.0342. This indicates that SPARK has good scalability, maintaining high prediction accuracy even when the model is reduced in size. This is due to the physics-guided compression and augmentation mechanisms in SPARK, which preserve key physical features while reducing complexity, ensuring stable model performance.

Table 2: Upstream pre-trained models of different sizes.

| MODEL SIZE | ERA5 | NS |
|---|---|---|
| 24.56MB | 0.0302 | 0.0723 |
| 17.65MB | 0.0323 | 0.0733 |
| 9.43MB | 0.0342 | 0.0734 |
| 4.57MB | 0.0388 | 0.0787 |
| 2.18MB | 0.0391 | 0.0798 |

The physical consistency and scalability of SPARK make it suitable for high-precision prediction of complex dynamical systems under limited computational resources.

### 4.5 TRANSFERABILITY OF SPARK (RQ4)

Table 3: Transfer the model pre-trained from full-data ERA5 to limited-data SEVIR. The results are presented in the formalization of $E \rightarrow S$, where $E$ is the model performance when it is trained from scratch and $S$ is the performance fine-tuned from the ERA5 pre-trained model.

| MSE | 20% SEVIR | 40% SEVIR | 60% SEVIR | 80% SEVIR | 100% SEVIR |
|---|---|---|---|---|---|
| SIMVP | 0.37→0.36 (-2.70%) | 0.36→0.34 (-5.56%) | 0.29→0.31 (+6.90%) | 0.25→0.26 (+4.00%) | 0.19→0.22 (+15.79%) |
| SIMVP + SPARK | 0.28→0.26 (-7.14%) | 0.27→0.24 (-11.11%) | 0.25→0.22 (-12.00%) | 0.21→0.19 (-9.52%) | 0.18→0.16 (-11.11%) |
| PREDRNN | 0.62→0.58 (-6.45%) | 0.52→0.41 (-21.15%) | 0.42→0.33 (-21.43%) | 0.27→0.24 (-11.11%) | 0.23→0.25 (+8.70%) |
| PREDRNN + SPARK | 0.30→0.27 (-10.00%) | 0.28→0.26 (-7.14%) | 0.27→0.24 (-11.11%) | 0.25→0.23 (-8.00%) | 0.22→0.19 (-13.64%) |
| EARTHFARSEER | 0.26→0.25 (-3.85%) | 0.24→0.24 (0.00%) | 0.23→0.21 (-8.70%) | 0.22→0.19 (-13.64%) | 0.16→0.17 (+6.25%) |
| EARTHFARSEER + SPARK | **0.24→0.22** (-8.33%) | **0.21→0.18** (-14.29%) | **0.19→0.17** (-10.53%) | **0.17→0.16** (-5.88%) | **0.15→0.13** (-13.33%) |

In the experimental design for transfer capability, we transfer the pretrained model from the full ERA5 dataset to the data-limited SEVIR dataset to evaluate SPARK's cross-domain transfer ability. The experiment is based on three baseline models (SimVP (Tan et al., 2022), PredRNN (Wang et al., 2017), Earthfarseer (Wu et al., 2024a)), comparing their performance with and without using SPARK. To better demonstrate the effectiveness of transfer learning, we pretrain on the full ERA5 dataset and then fine-tune on different amounts of SEVIR dataset (20%, 40%, 60%, 80%, 100%), using mean squared error (MSE x 100) as the evaluation metric. Results as shown in Tabel 3, we have two key observations as follows:

**Obs 1: SPARK significantly improves transfer performance.** For all baseline models, using SPARK for fine-tuning significantly reduces the error on the SEVIR dataset. For instance, Earthfarseer's error decreases from 0.21 to 0.18 (a reduction of 14.29%) after pretraining on 40% of

ERA5 data, which demonstrate a significant advantage over training without SPARK (error remains unchanged). This indicates that SPARK effectively leverages physics-guided information to enhance the model's transfer learning capability.

**Obs 2: SPARK performs especially well with limited data.** With smaller data amounts (e.g., 20% of ERA5 data), models using SPARK show greater performance improvements. For SimVP-V2, when pretrained with 20% of the data, MSE decreases from 0.28 to 0.26 (a reduction of 7.14%), while without SPARK, the reduction is only 2.70%. This result demonstrates that SPARK exhibits strong transfer capabilities, particularly in data-scarce scenarios, effectively addressing data insufficiency in cross-domain environments and significantly enhancing model generalization.

## 4.6 COMPARISON OF OOD HANDLING BETWEEN SPARK AND OTHER METHODS

To highlight the capability of our model in handling OOD scenarios, we further select state-of-the-art OOD baselines for comparison. In particular, we include LEADS (Kirchmeyer et al., 2022), CODA (Yin et al., 2021) and NUWA (Wang et al., 2024b) in our experiments. These are selected

Table 4: Performance comparison of different models with and without OOD conditions on the Prometheus, ERA5, and Spherical-SWE benchmark datasets. SPARK achieves the best results in all scenarios, showing stronger generalization and robustness.

| | BENCHMARKS | | | | | |
|---|---|---|---|---|---|---|
| MODEL | PROMETHEUS | | ERA5 | | SPHERICAL-SWE | |
| | *w/o* OOD | *w/* OOD | *w/o* OOD | *w/* OOD | *w/o* OOD | *w/* OOD |
| LEADS 2022 | 0.0374 | 0.0403 | 0.2367 | 0.4233 | 0.0038 | 0.0047 |
| CODA 2021 | 0.0353 | 0.0372 | 0.1233 | 0.2367 | 0.0034 | 0.0043 |
| NUWA 2024B | 0.0359 | 0.0398 | 0.0645 | 0.0987 | 0.0032 | 0.0039 |
| SPARK (OURS) | **0.0323** | **0.0328** | **0.0398** | **0.0401** | **0.0022** | **0.0024** |

due to their relevance in tackling dynamics modeling and boundary condition incorporation, which aligns with our focus on fluid dynamics modeling under different input conditions. Experimental results on three benchmark datasets are illustrated in Table 4.

The results from Table 4 show that SPARK performs significantly better in handling OOD scenarios, especially on the Prometheus, ERA5, and Spherical-SWE benchmarks. On the Prometheus dataset, SPARK achieves errors of 0.0323 and 0.0328 for w/o OOD and w/ OOD, respectively, clearly outperforming LEADS (0.0374 and 0.0403), CODA (0.0353 and 0.0372), and NUWA (0.0359 and 0.0398), showing strong generalization and adaptability to changes. On the ERA5 dataset, SPARK's errors are 0.0398 and 0.0401, while other models show significantly increased errors under OOD, such as LEADS increasing from 0.2367 to 0.4233, indicating better robustness of SPARK under complex atmospheric conditions. On the Spherical-SWE dataset, SPARK achieves errors of 0.0022 and 0.0024 for w/o OOD and w/ OOD settings, respectively, compared to errors of 0.0038 and 0.0047 by models like LEADS. This further demonstrates SPARK's stability and accuracy in handling boundary condition changes and complex spatiotemporal systems. The results clearly demonstrate that SPARK has excellent performance, high robustness, and strong generalization in OOD scenarios.

## 5 CONCLUSION

In this paper, we propose SPARK, a novel physics-guided quantitative augmentation framework aimed at improving out-of-distribution generalization on dynamical system modeling. SPARK first construct a discrete memorybank for efficient augmentation by incorporating boundary information and physical parameters through a vector quantization-based compression mechanism. Additionally, SPARK integrates a fourier-enhanced graph ODE for efficient and precise system prediction, further addressing temporal distribution shifts. Experimental results on both real-world and synthetic datasets demonstrate that SPARK outperforms existing methods under distribution shifts.

## ETHICS STATEMENT

We acknowledge that all co-authors of this work have read and committed to adhering to the ICLR Code of Ethics.

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

# A  PROOFS OF THEOREM 1

We prove that introducing physical priors improves model generalization from an information-theoretic perspective, based on the relationship between mutual information and generalization error.

## A.1  PRELIMINARIES

### A.1.1  GENERALIZATION ERROR AND EMPIRICAL ERROR

▷ True Risk (Expected Loss):

$$\mathcal{L}(\theta) = \mathbb{E}_{(X,Y)\sim\mathcal{D}_{\text{true}}}[\ell(\theta; X, Y)], \tag{14}$$

where $\ell(\theta; X, Y)$ is the loss function, and $\mathcal{D}_{\text{true}}$ is the true data distribution.

▷ Empirical Risk:

$$\mathcal{L}_{\text{emp}}(\theta; \mathcal{D}) = \frac{1}{n}\sum_{i=1}^{n}\ell(\theta; x_i, y_i), \tag{15}$$

where $\mathcal{D} = \{(x_i, y_i)\}_{i=1}^{n}$ is the training dataset.

### A.1.2  MUTUAL INFORMATION

▷ Mutual Information:

$$I(\theta; \mathcal{D}) = \mathbb{E}_{\theta,\mathcal{D}}\left[\log\frac{p(\theta, \mathcal{D})}{p(\theta)p(\mathcal{D})}\right]. \tag{16}$$

▷ Conditional Mutual Information (given physical prior $\mathcal{P}$):

$$I(\theta; \mathcal{D} \mid \mathcal{P}) = \mathbb{E}_{\theta,\mathcal{D},\mathcal{P}}\left[\log\frac{p(\theta, \mathcal{D} \mid \mathcal{P})}{p(\theta \mid \mathcal{P})p(\mathcal{D} \mid \mathcal{P})}\right]. \tag{17}$$

## A.2  PROOF

We then prove that introducing physical prior information reduces the upper bound of the generalization error.

**Relating Generalization Error to Mutual Information**

According to information-theoretic results[1], for any learning algorithm, the expected generalization error has the following upper bound:

$$\left|\mathbb{E}_{\theta,\mathcal{D}}\left[\mathcal{L}(\theta) - \mathcal{L}_{\text{emp}}(\theta; \mathcal{D})\right]\right| \leq \sqrt{\frac{2I(\theta; \mathcal{D})}{n}}. \tag{18}$$

**Introducing Physical Prior Information**

When we introduce physical prior information $\mathcal{P}$, we consider the conditional mutual information $I(\theta; \mathcal{D} \mid \mathcal{P})$. Since $\mathcal{P}$ is known, we can reconsider the upper bound on the generalization error under the condition of $\mathcal{P}$.

**Recalculate the Upper Bound of Generalization Error**

Based on conditional mutual information, the upper bound becomes:

$$\left|\mathbb{E}_{\theta,\mathcal{D},\mathcal{P}}\left[\mathcal{L}(\theta) - \mathcal{L}_{\text{emp}}(\theta; \mathcal{D}) \mid \mathcal{P}\right]\right| \leq \sqrt{\frac{2I(\theta; \mathcal{D} \mid \mathcal{P})}{n}}. \tag{19}$$

---

[1]Reference: Xu, A., & Raginsky, M. (2017). Information-theoretic analysis of generalization capability of learning algorithms. *Advances in Neural Information Processing Systems*, 30.

Since $\mathcal{P}$ is fixed, we can take the expectation over $\mathcal{P}$:

$$\left| \mathbb{E}_{\theta,\mathcal{D}} \left[ \mathcal{L}(\theta) - \mathcal{L}_{\text{emp}}(\theta;\mathcal{D}) \right] \right| \leq \sqrt{\frac{2I(\theta;\mathcal{D} \mid \mathcal{P})}{n}}. \tag{20}$$

**Physical Prior Reduces Mutual Information**

The physical prior $\mathcal{P}$ provides additional knowledge about the model parameters $\theta$, which reduces the mutual information $I(\theta;\mathcal{D} \mid \mathcal{P})$ between $\theta$ and $\mathcal{D}$ under the condition $\mathcal{P}$.

Intuitively, the physical prior restricts the possible parameter space, reducing the influence of training data on parameters and thus decreasing the model's dependence on the training data. Mathematically, mutual information satisfies:

$$I(\theta;\mathcal{D}) \geq I(\theta;\mathcal{D} \mid \mathcal{P}). \tag{21}$$

Combining the above steps, we conclude:

$$\left| \mathbb{E}_{\theta,\mathcal{D}} \left[ \mathcal{L}(\theta) - \mathcal{L}_{\text{emp}}(\theta;\mathcal{D}) \right] \right| \leq \sqrt{\frac{2I(\theta;\mathcal{D} \mid \mathcal{P})}{n}} \leq \sqrt{\frac{2I(\theta;\mathcal{D})}{n}}. \tag{22}$$

Thus, introducing physical prior information $\mathcal{P}$ reduces the mutual information $I(\theta;\mathcal{D} \mid \mathcal{P})$, leading to a reduced upper bound on generalization error, and thereby improving the model's generalization capability.

### A.3 Generalization Error Bound in Bayesian Learning with Physical Prior

Consider a hypothesis space $\mathcal{H}$, with model parameters $\theta \in \mathcal{H}$, a training dataset $\mathcal{D} = \{(x_i, y_i)\}_{i=1}^n$, and a loss function $\ell(\theta; x, y)$. The true risk is:

$$\mathcal{L}(\theta) = \mathbb{E}_{(x,y)\sim\mathcal{P}_{\text{data}}}[\ell(\theta; x, y)], \tag{23}$$

and the empirical risk is:

$$\mathcal{L}_{\text{emp}}(\theta) = \frac{1}{n} \sum_{i=1}^n \ell(\theta; x_i, y_i). \tag{24}$$

Assume the prior distribution $P(\theta)$ contains physical prior information, and the posterior distribution is $Q(\theta)$. For any $\delta > 0$, with probability at least $1 - \delta$, the generalization error has the following upper bound:

$$\mathbb{E}_{\theta\sim Q}[\mathcal{L}(\theta)] \leq \mathbb{E}_{\theta\sim Q}[\mathcal{L}_{\text{emp}}(\theta)] + \sqrt{\frac{KL(Q\|P) + \ln\dfrac{2\sqrt{n}}{\delta}}{2n}}, \tag{25}$$

where $KL(Q\|P)$ is the Kullback-Leibler divergence between the posterior $Q$ and the prior $P$.

Introducing physical prior information as the prior distribution $P(\theta)$ reduces the Kullback-Leibler divergence $KL(Q\|P)$ between the posterior $Q(\theta)$ and the prior $P(\theta)$. According to the theorem, this reduces the upper bound of the generalization error. This implies that, within the Bayesian learning framework, incorporating physical prior information enhances the model's generalization ability, leading to better performance on unseen data.

## B Proofs of Theorem 2

We prove that introducing physical priors improves model generalization from a Bayesian learning perspective, using the PAC-Bayesian theory. The PAC-Bayesian theorem provides an upper bound on the generalization error for randomized algorithms, which relates to the KL divergence between prior and posterior distributions.

## B.1 PRELIMINARIES

▷ PAC-Bayesian Theorem

The PAC-Bayesian theorem provides a probabilistic upper bound on the generalization performance of randomized learning algorithms. The core idea is to define a probability distribution over the hypothesis space and use the KL divergence between the prior and posterior to quantify generalization error.

▷ KL Divergence (Relative Entropy)

For two probability distributions $P$ and $Q$, the KL divergence is defined as:

$$KL(Q\|P) = \int \ln\left(\frac{dQ}{dP}\right) dQ. \tag{26}$$

The KL divergence measures how far the distribution $Q$ deviates from $P$.

## B.2 PROOF

**Define the Randomized Prediction Function**

In the Bayesian framework, the model parameter $\theta$ is treated as a random variable, whose distribution is given by the posterior distribution $Q(\theta)$. During prediction, the model samples $\theta$ from the posterior and uses it for prediction.

**Introduce Physical Prior Information**

Physical prior information is encoded in the prior distribution $P(\theta)$. This prior reflects our belief about the model parameters before observing data.

**Apply the PAC-Bayesian Theorem**

According to the PAC-Bayesian theorem, for any posterior distribution $Q(\theta)$, with probability at least $1 - \delta$, we have:

$$\mathbb{E}_{\theta \sim Q}[\mathcal{L}(\theta)] \leq \mathbb{E}_{\theta \sim Q}[\mathcal{L}_{\text{emp}}(\theta)] + \sqrt{\frac{KL(Q\|P) + \ln\frac{2\sqrt{n}}{\delta}}{2n}}. \tag{27}$$

**Note**: The full proof of this theorem involves the Hoeffding inequality and variations of martingale inequalities, but here we focus on applying the conclusion.

**Interpret the Role of KL Divergence**

The smaller the KL divergence $KL(Q\|P)$, the closer the posterior distribution $Q$ is to the prior distribution $P$. This means that the model deviates less from the prior information during learning.

Introducing physical prior information makes the prior distribution $P(\theta)$ closer to the true parameter distribution, reducing the KL divergence between the posterior $Q(\theta)$ and prior $P(\theta)$, i.e., $KL(Q\|P)$ decreases.

**Derive the Reduction in Generalization Error Bound**

Since $KL(Q\|P)$ decreases, the PAC-Bayesian upper bound on the generalization error also decreases:

$$\mathbb{E}_{\theta \sim Q}[\mathcal{L}(\theta)] - \mathbb{E}_{\theta \sim Q}[\mathcal{L}_{\text{emp}}(\theta)] \leq \sqrt{\frac{\downarrow KL(Q\|P) + \ln\frac{2\sqrt{n}}{\delta}}{2n}}. \tag{28}$$

Thus, introducing physical prior information reduces the upper bound on the generalization error, thereby improving the model's generalization capability.

### B.3 DISCUSSION

#### B.3.1 ROLE OF PHYSICAL PRIOR INFORMATION

▷ **Narrowing the Parameter Space**: Physical priors restrict the possible values of model parameters, making the prior distribution $P(\theta)$ more concentrated in regions that follow physical laws. ▷ **Guiding the Posterior Distribution**: Since the prior distribution includes physical information, the posterior distribution tends to favor parameter regions that are consistent with physical laws during the update.

#### B.3.2 RELATIONSHIP BETWEEN KL DIVERGENCE AND GENERALIZATION ERROR

▷ **KL Divergence as a Measure of Deviation**: KL divergence measures how much the posterior deviates from the prior. The smaller the deviation, the lower the upper bound on the generalization error. ▷ **Coordination between Prior and Posterior**: A good prior allows the model to make less drastic adjustments to parameters given the data, thereby maintaining model stability and generalizability.

#### B.3.3 ADVANTAGES OF THE BAYESIAN LEARNING FRAMEWORK

▷ **Naturally Incorporates Prior Knowledge**: The Bayesian approach allows prior knowledge to be incorporated into the model as a probability distribution, which helps improve model performance, especially when data is limited. ▷ **Probabilistic Interpretation**: The PAC-Bayesian theorem provides an upper bound on the generalization error with probabilistic guarantees, making the theoretical results more robust.

## C THE PROPOSED SPARK ALGORITHM

The whole learning algorithm of SPARK is summarized in Algorithm 1.

---

**Algorithm 1** Training of SPARK

---

**Require:** historical observations $\{\mathbf{s}_i^{1:T_0}\}_{i=1}^N$; boundary information $\mathfrak{B}$; physical parameter set $\boldsymbol{\lambda}$
1: **Stage 1: Data Compression with Physical Priors**
2: Randomly initialize parameters of memorybank $\Theta_{\mathcal{M}}$, GNN encoder $\Theta_E$, linear decoder $\Theta_{\mathcal{I}}$
3: **for** each sensor $i$ **do**
4:     Incorporate boundary information $\mathfrak{B}$ into observations $\boldsymbol{s}_i$ w.r.t position embeddings
5:     Fuse physical parameter $\boldsymbol{\lambda}$ through designed channel attention $\boldsymbol{a}_\vartheta$
6:     Implement vector quantization with GNN encoder $\Theta_E$ and discrete memorybank $\Theta_{\mathcal{M}}$
7:     Fed discretized representations $\boldsymbol{z}$ into linear decoder $\mathcal{I}$ for reconstruction
8: **end for**
9: Optimize the pre-training framework with memorybank $\mathcal{M}$
10: **Stage 2: Memorybank-Guided Data Augmentation**
11: Sample instances from training set with probabilistic model
12: **for** each instance $\mathcal{X}_i$ **do**
13:     Search Top-K nearest discrete embeddings $\{\boldsymbol{e}_1, \cdots, \boldsymbol{e}_K\}$ for augmentation
14: **end for**
15: **Stage 3: Fourier-enhanced Graph ODE**
16: **for** each observation sequence $\{\mathcal{X}_i, \mathcal{Y}_i\}$ **do**
17:     Initialize the observation state with attention mechanism
18:     Employ attention mechanism to map historical observations $\mathcal{X}_i$ into hidden state $\boldsymbol{h}_i$
19:     model hidden state through fourier-enhanced graph ODE $\Phi(\cdot)$
20:     Predict future observations $\mathcal{Y}_i$
21: **end for**
22: Optimize framework by minimizing the MSE loss

---

## D  DETAILED DESCRIPTION OF DATASETS

We evaluate our proposed SPARK on benchmark datasets in three fields: *Prometheus* for computational fluid dynamics; *ERA5* for real-world scenarios; *2D Navier-Stokes Equations*, *Spherical Shallow Water Equations*, and *3D Reaction-Diffusion Equations* for partial differential equations.

*Prometheus* (Wu et al., 2024b) is a large-scale, out-of-distribution (OOD) fluid dynamics dataset designed for the development and benchmarking of machine learning models, particularly those that predict fluid dynamics under varying environmental conditions. This dataset includes simulations of tunnel and pool fires (represented as Prometheus-T and Prometheus-P in experiments), encompassing a wide range of fire dynamics scenarios modeled using fire dynamics simulators that solve the Navier-Stokes equations. Key features of the dataset include 25 different environmental settings with variations in parameters such as Heat Release Rate (HRR) and ventilation speeds. In total, the Prometheus dataset encompasses 4.8 TB of raw data, which is compressed to 340 GB. It not only enhances the research on fluid dynamics modeling but also aids in the development of models capable of handling complex, real-world scenarios in safety-critical applications like fire safety management and emergency response planning.

*ERA5* (Hersbach et al., 2020) is a global atmospheric reanalysis dataset developed by the European Centre for Medium-Range Weather Forecasts (ECMWF), offering comprehensive weather data from 1979 to the present with exceptional spatial resolution (31 km) and hourly temporal granularity. This dataset encompasses a rich array of meteorological variables, including but not limited to surface pressure, sea surface temperature, sea surface height, and two-meter air temperature. *ERA5* is extensively employed across a multitude of domains, including climate modeling, environmental monitoring, atmospheric dynamics research, and energy management optimization. The dataset's integration of physical models with vast observational data makes it a cornerstone for advancing predictive models in meteorology and climate science.

*2D Navier-Stokes Equations* (Li et al., 2021) describe the motion of fluid substances such as liquids and gases. These equations are a set of partial differential equations that predict weather, ocean currents, water flow in a pipe, and air flow around a wing, among other phenomena. The equations arise from applying Newton's second law to fluid motion, together with the assumption that the fluid stress is the sum of a diffusing viscous term proportional to the gradient of velocity, and a pressure term. The equations are expressed as follows:

$$\rho \left( \frac{\partial u}{\partial t} + u \cdot \nabla u \right) = -\nabla p + \nabla \cdot \tau + f,$$

$$\frac{\partial \rho}{\partial t} + \nabla \cdot (\rho u) = 0, \tag{29}$$

$$\frac{\partial (\rho E)}{\partial t} + \nabla \cdot ((\rho E + p)u) = \nabla \cdot (\tau \cdot u) + \nabla \cdot (k \nabla T) + \rho f \cdot u,$$

where $u$ denotes the velocity field, $\rho$ represents the density of the fluid, $p$ is the pressure, $\tau$ is the viscous stress tensor, given by $\mu(\nabla u + (\nabla u)^T) - \frac{2}{3}\mu(\nabla \cdot u)\mathbf{I}$. $E$ is the total energy per unit mass, $E = e + \frac{1}{2}|u|^2$, $e$ is the internal energy per unit mass, $T$ denotes the temperature, and $k$ represents the thermal conductivity.

*Spherical Shallow Water Equations* (Galewsky et al., 2004) model surface water flows under the assumption of a shallow depth compared to horizontal dimensions. This simplification leads to the Shallow Water equations, a set of partial differential equations (PDEs) that describe the flow below a pressure surface in a fluid (often water). Shallow Water typically encompasses variables such as water surface elevation and the two components of velocity field ($u$-velocity in the $x$-direction and $v$-velocity in the $y$-direction). These properties are crucial for modeling waves, tides, and large-scale circulations in oceans and atmospheres. The equations consist of a continuity equation for mass conservation and a momentum equation for momentum conservation:

$$\frac{\partial h}{\partial t} + \nabla \cdot (h \cdot u) = 0,$$

$$\frac{\partial u}{\partial t} + (u \cdot \nabla)u + g \nabla h = 0, \tag{30}$$

where $h$ represents the fluid depth, $u$ is the velocity field, and $g$ denotes the acceleration due to gravity. These equations are used extensively in environmental modeling, including weather forecasting, oceanography, and climate studies.

*3D Reaction-Diffusion Equations* (Rao et al., 2023) are a class of partial differential equations (PDEs) that describe the temporal and spatial evolution of chemical species in three-dimensional domains. These equations are fundamental for modeling systems where chemical substances not only react but also diffuse through a 3D medium. The interaction between reaction kinetics and diffusion mechanisms leads to intricate spatiotemporal dynamics that are critical in fields such as biology, chemistry, and physics. The general form of the 3D reaction-diffusion system can be expressed as:

$$
\begin{aligned}
\frac{\partial u}{\partial t} &= D_u \nabla^2 u + f(u,v,w) - g(u,v)u + \alpha_u + \sigma_u S_u(x,y,z,t), \\
\frac{\partial v}{\partial t} &= D_v \nabla^2 v + h(u,v)u^2 - \beta v + \alpha_v + \sigma_v S_v(x,y,z,t), \\
\frac{\partial w}{\partial t} &= D_w \nabla^2 w + p(v,w) - \gamma w + \alpha_w + \sigma_w S_w(x,y,z,t),
\end{aligned}
\tag{31}
$$

where $u$, $v$, and $w$ represent the concentrations of different chemical species, $D_u$, $D_v$, and $D_w$ denote their respective diffusion coefficients. The terms $f(u,v,w)$ and $p(v,w)$ describe the reaction kinetics that govern the interactions between these species, while $g(u,v)$ and $h(u,v)$ control the rate of conversion and interaction. $\alpha_u$, $\alpha_v$, and $\alpha_w$ represent constant growth rates, and $\sigma_u$, $\sigma_v$, and $\sigma_w$ introduce noise terms that model stochastic external influences through spatially and temporally dependent source functions $S_u(x,y,z,t)$, $S_v(x,y,z,t)$, and $S_w(x,y,z,t)$. The resulting system of equations provides a robust framework for simulating complex dynamical behaviors in three-dimensional reactive-diffusive environments.

## E   DETAILS OF COMPARED APPROACHES

The approaches compared in this study are listed as follows:

- **U-Net** Ronneberger et al. (2015) is a convolutional neural network originally developed for biomedical image segmentation. Its U-shaped architecture with symmetric skip connections between the encoder and decoder facilitates effective feature integration.
- **ResNet** He et al. (2016) introduces residual connections to address the issue of performance degradation in deep networks. These skip connections allow information to bypass layers, enabling deeper and more trainable architectures.
- **ViT** Dosovitskiy et al. (2021) utilizes the Transformer model for image classification. The image is divided into patches, which are processed using self-attention mechanisms, achieving a balance between computational efficiency and accuracy.
- **SwinT** Liu et al. (2021b) employs a sliding window technique for the extraction of both local and global features. This makes it versatile for a wide range of computer vision tasks.
- **FNO** Li et al. (2021) leverages Fourier transforms for extracting global features, making it particularly effective for handling continuous field data and solving partial differential equations (PDEs).
- **UNO** Ashiqur Rahman et al. (2022) combines U-Net's architecture with optimization techniques to boost feature extraction and fusion, thereby enhancing the model's overall performance.
- **CNO** Raonic et al. (2024) integrates convolutional operations with operator learning to better handle high-dimensional continuous data, focusing on the modeling of intricate dynamic systems.
- **NMO** Wu et al. (2024c) improves the capability to model multi-scale dynamic systems by integrating neural networks with manifold learning methods.

## F   METRICS DETAILS

**Mean Squared Error (MSE)**: This metric provides the average of the squares of the differences between the actual and predicted values. A lower MSE indicates a closer fit of the predictions to the

true values. It's given by the equation:

$$\text{MSE} = \frac{1}{N} \sum_{i=1}^{N} (V_{\text{true},i} - V_{\text{fut},i})^2 \tag{32}$$

where $V_{\text{true},i}$ represents the true value, $V_{\text{fut},i}$ denotes the predicted value, and $N$ is the number of observations.

**Multi-Scale Structural Similarity (SSIM)**: SSIM is designed to provide an assessment of the structural integrity and similarity between two images, $x$ and $y$. Higher SSIM values suggest that the structures of the two images being compared are more similar.

$$\text{SSIM}(x, y) = \frac{(2\mu_x \mu_y + c_1)(2\sigma_{xy} + c_2)}{(\mu_x^2 + \mu_y^2 + c_1)(\sigma_x^2 + \sigma_y^2 + c_2)} \tag{33}$$

where $\mu$ is the mean, $\sigma$ represents variance, and $c_1$ and $c_2$ are constants to avoid instability.

**Peak Signal-to-Noise Ratio (PSNR)**: PSNR gauges the quality of a reconstructed image compared to its original by measuring the ratio between the maximum possible power of the signal and the power of corrupting noise. A higher PSNR indicates a better reconstruction quality.

$$\text{PSNR} = 10 \times \log_{10} \left( \frac{\text{MAX}_I^2}{\text{MSE}} \right) \tag{34}$$

where $\text{MAX}_I$ is the maximum possible pixel value of the image.

## G  PARAMETER SENSITIVITY ANALYSIS

To investigate the influence of hyperparameter $k$, we add experiments on the value of $k$ on the Navier-Stokes, Prometheus, 3D Reaction–Diff, and ERA5 datasets. The candidate values are $\{1, 3, 5, 7, 9, 11\}$, and the results are shown in Table 5.

Table 5: Performance comparison of different $k$.

| k | Navier–Stokes | Spherical-SWE | Prometheus | 3D Reaction–Diff |
|---|---|---|---|---|
| 1 | 0.0752 | 0.0022 | 0.0315 | 0.0116 |
| 3 | 0.0726 | **0.0018** | **0.0296** | 0.0108 |
| 5 | **0.0715** | 0.0021 | 0.0303 | **0.0104** |
| 7 | 0.0731 | 0.0024 | 0.0311 | 0.0110 |
| 9 | 0.0764 | 0.0025 | 0.0320 | 0.0121 |
| 11 | 0.0780 | 0.0029 | 0.0327 | 0.0128 |

As $k$ increases, the model's performance first improves and then declines, with optimal performance generally achieved when $k$ is between 3 and 5.

## H  EXTERNAL EXPERIMENTS

### H.1  ABLATION STUDY

To further demonstrate the contribution of each strategy, we conduct ablation experiments with five model variants. The experiments are conducted on Prometheus and Navier–Stokes datasets with OOD scenarios, and the results are shown in Table 6.

Table 6: Performance comparison of different model variants.

| | Ours | w/o param | w/o bound | w/o param&bound | w/o MemBank | w/o CL |
|---|---|---|---|---|---|---|
| Prometheus | **0.0301** | 0.0357 | 0.0324 | 0.0397 | 0.0416 | 0.0338 |
| Navier–Stokes | **0.0725** | 0.0833 | 0.0764 | 0.0902 | 0.1058 | 0.0792 |

As observed, removing physical parameters or boundary conditions during pretraining leads to a performance decline, with an even greater drop when the memory bank is discarded. This validates the effectiveness of physical compression when addressing OOD problems.

## H.2  Low-data Regime Experiments

To explore the performance of our model in very low-data regime of transfer learning, we conduct experiments here. Specifically, after pre-training on the full ERA5 dataset, we finetune on subsets of the Sevir dataset with varying amounts of data (1%, 3%, 5%, and 10%). The detailed comparison of baseline models (PredRNN and SimVP) with and without the SPARK plugin is shown in Table 7.

Table 7: Performance comparison of varying data amounts.

|  | 1% Sevir | 3% Sevir | 5% Sevir | 10% Sevir |
|---|---|---|---|---|
| PredRNN | 3.51→3.38 | 2.57→2.35 | 1.83→1.68 | 1.22→1.16 |
| PredRNN+SPARK | 3.37→3.02 | 2.49→2.14 | 1.72→1.45 | 1.14→0.97 |
| SimVP | 2.43→2.20 | 1.86→1.55 | 1.29→1.11 | 0.75→0.68 |
| SimVP+SPARK | 2.30→1.98 | 1.75→1.23 | 1.21→0.98 | 0.71→0.57 |

The results show that models with SPARK plugin consistently outperform their baseline models in the very low data regime.

## H.3  Experiments on 1-D data

To explore model performance on 1-D data, we add experiments on 1-D data using the Burgers Equations. The results are shown in Table 8, which indicate that our method is also applicable to 1-D data.

Table 8: Performance on 1-D Burgers Equations.

|  | U-Net | ResNet | FNO | CNO | NMO | Ours+SPARK |
|---|---|---|---|---|---|---|
| **w/o OOD** | 0.362 | 0.338 | 0.298 | 0.314 | 0.246 | 0.228 |
| **w/ OOD** | 0.397 | 0.351 | 0.325 | 0.338 | 0.273 | 0.243 |

## H.4  Cross-dimension Generalization Experiments

We select FNO, CNO, and NMO as baselines to evaluate SPARK's generalization capability across different dimensional data. Specifically, we pre-train on 2-D Navier-Stokes Equations and finetune on 1-D Burgers Equations. The results shown in Table 9 validate that model variants with SPARK plugin have better generalization capability than their baseline models.

Table 9: Cross-dimension generalization performance.

|  | FNO | FNO+SPARK | CNO | CNO+SPARK | NMO | NMO+SPARK |
|---|---|---|---|---|---|---|
| **Burgers** | 0.317→0.294 | 0.308→0.275 | 0.298→0.275 | 0.280→0.256 | 0.241→0.223 | 0.228→0.204 |

## H.5  Challenging Task Experiments

We add two challenging experiments, namely long-term prediction and extreme event prediction. We choose Prometheus and Sevir datasets to conduct the two experiments, separately.

**long-term prediction.**

For long-term prediction, we use Prometheus with ten steps as input and supervise the prediction of the next ten steps during training. During inference, we predict the next 10, 30, and 50 steps in an autoregressive manner. The results in Table 10 demonstrate that our model outperforms other baselines in long-term prediction performance.

**Extreme event prediction.**

For extreme event prediction, we use Sevir dataset, which contains data related to severe weather phenomena. To better evaluate the prediction performance of extreme events, we used the Critical

Table 10: Long-term prediction performance.

| Time Step | U-Net | ViT | FNO | NMO | Ours |
|---|---|---|---|---|---|
| 10 | 0.0931 | 0.0674 | 0.0447 | 0.0397 | **0.0294** |
| 30 | 0.1374 | 0.1038 | 0.0815 | 0.0726 | **0.0537** |
| 50 | 0.2238 | 0.1842 | 0.1374 | 0.1154 | **0.0921** |

Success Index (CSI), in addition to MSE. For simplicity, we used only the thresholds 16, 133, 181, 219 and the mean CSI-M. The results in Table 11 show that our model consistently outperform these baselines in extreme event prediction.

Table 11: Extreme event prediction performance.

| Model | CSI-M ↑ | CSI-219 ↑ | CSI-181 ↑ | CSI-133 ↑ | CSI-16 ↑ | MSE ($10^{-3}$) ↓ |
|---|---|---|---|---|---|---|
| U-Net | 0.3593 | 0.0577 | 0.1580 | 0.3274 | 0.7441 | 4.1119 |
| ViT | 0.3692 | 0.0965 | 0.1892 | 0.3465 | 0.7326 | 4.1661 |
| PredRNN | 0.4028 | 0.1274 | 0.2324 | 0.3858 | 0.7507 | 3.9014 |
| SimVP | 0.4275 | 0.1492 | 0.2538 | 0.4084 | 0.7566 | 3.8182 |
| Ours | **0.4683** | **0.1721** | **0.2734** | **0.4375** | **0.7792** | **3.6537** |

## H.6 COMPARISON OF TRAINING COST

We add experiments of computational costs on Prometheus dataset. To be fair, we conduct the experiments on a single NVIDIA 40GB A100 GPU. From the Table 12, we can observe that our method has a competitive computation cost.

Table 12: Comparison performance of training cost.

| Method | UNet | ResNet | VIT | SwinT | FNO | UNO | CNO | NMO | Ours |
|---|---|---|---|---|---|---|---|---|---|
| **Training time (h)** | 11.2 | 9.76 | 14.5 | 12.3 | 6.9 | 7.8 | 13.4 | 6.3 | 6.7 |
| **Inference time (s)** | 1.34 | 0.93 | 1.32 | 1.13 | 0.54 | 0.67 | 0.12 | 0.52 | 0.55 |

## H.7 COMPARISON WITH OOD-SPECIFIC MODELS

Our SPARK is specifically designed for OOD problem. Here, we use three models specialized in OOD dynamical system modeling mentioned before, along with FNO, for comparison. The results shown in Table 13 indicate that OOD-specific models outperform FNO in both OOD and non-OOD scenarios, with SPARK achieving the best performance.

Table 13: Comparison performance with OOD-specific models.

| Dataset | Prometheus (ID) | Prometheus (OOD) | ERA5 (ID) | ERA5 (OOD) | SSWE (ID) | SSWE (OOD) |
|---|---|---|---|---|---|---|
| FNO | 0.0547 | 0.0606 | 0.7233 | 0.9821 | 0.0061 | 0.0084 |
| LEADS | 0.0374 | 0.0403 | 0.2367 | 0.4233 | 0.0038 | 0.0047 |
| CODA | 0.0353 | 0.0372 | 0.1233 | 0.2367 | 0.0034 | 0.0043 |
| NUWA | 0.0359 | 0.0398 | 0.0645 | 0.0987 | 0.0032 | 0.039 |
| Ours | **0.0323** | **0.0328** | **0.0398** | **0.0401** | **0.0022** | **0.0024** |

## H.8 COMPARISON WITH DGODE

We run DGODE's open-source code and conduct comparative experiments in both non-OOD (ID) and OOD scenarios. The results shown in Table 14 indicate that SPARK performs better.

Table 14: Comparison performance with DGODE.

| Dataset | Prometheus (ID) | Prometheus (OOD) | ERA5 (ID) | ERA5 (OOD) | SSWE (ID) | SSWE (OOD) |
|---------|-----------------|------------------|-----------|------------|-----------|------------|
| **DGODE** | 0.0344 | 0.0359 | **0.0387** | 0.0435 | 0.0024 | 0.0029 |
| **Ours** | **0.0323** | **0.0328** | 0.0398 | **0.0401** | **0.0022** | **0.0024** |

## H.9 COMPARISON WITH BEAMVQ

To, compare with BeamVQ, we conduct experiments on the Navier–Stokes, Spherical-SWE, Prometheus, and 3D Reaction–Diff dataset. Here, we use FNO and SimVP as backbones. Further, we select parameter count, training time, and inference time on Navier–Stokes to compare the effiency of two models. Table 15 and Table 16 below shows that SPARK is much more lightweight and performs better. Notably, the SimVP+BeamVQ model variant crashes on 3D Reaction-Diff due to memory overflow, as its parameter complexity is unsuitable for 3D scenarios.

Table 15: Comparison performance with BeamVQ.

| Method | Navier–Stokes | SSWE | Prometheus | 3D Reaction–Diff |
|--------|---------------|------|------------|------------------|
| FNO | 0.1556 | 0.0038 | 0.0447 | 0.0132 |
| FNO+BeamVQ | 0.1342 | 0.0032 | 0.0356 | 0.0104 |
| FNO+SPARK | 0.1257 | 0.0029 | 0.0338 | 0.0095 |
| SimVP | 0.1262 | 0.0031 | 0.0394 | 0.0108 |
| SimVP+BeamVQ | 0.1173 | 0.0027 | 0.0375 | - |
| SimVP+SPARK | 0.1105 | 0.0024 | 0.0360 | 0.0087 |

Table 16: Effiency performance with BeamVQ.

| Method | MSE | Param | Training Time | Inference Time |
|--------|-----|-------|---------------|----------------|
| FNO+BeamVQ | 0.1342 | 214.25 MB | 26.11 h | 3.25 s |
| FNO+SPARK | 0.1257 | 35.67 MB | 4.2 h | 0.58 s |

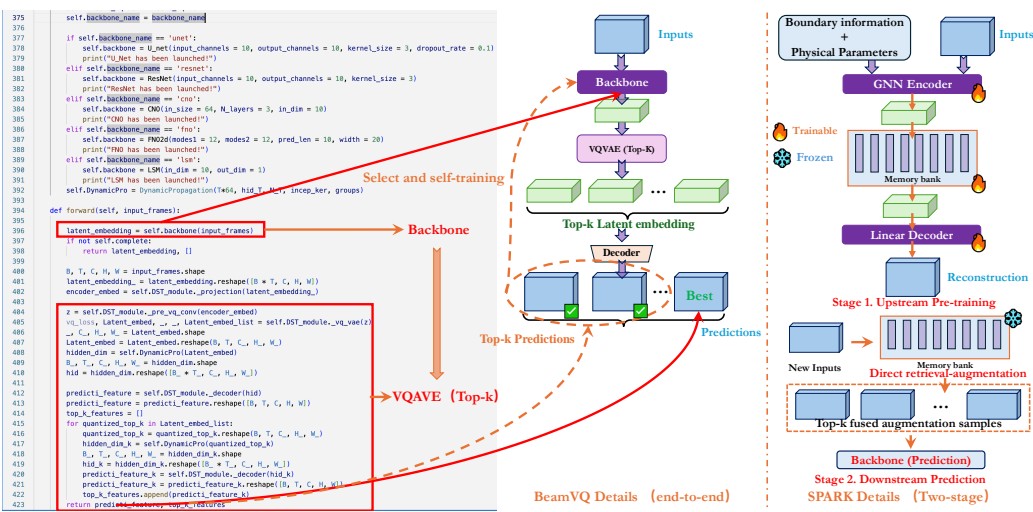

Figure 7: Schematic comparison of SPARK and BeamVQ.

# I DETAILS OF MODEL ARCHITECTURE AND RELATED SETUP

Table 17: Details of SPARK's upstream architecture.

| Upstream | | |
|---|---|---|
| **Procedure** | **Layer** | **Dimension** |
| Boundary information injection | Boundary Fusion (Concat + Linear) | (4096, 128) |
| | Boundary Encoding (Linear + LayerNorm) | (4096, 128) |
| Physical parameters injection | Channel attention | (2, 128) |
| | Aggregation | (4096, 128) |
| GNN reconstruction | Graph Encoder (GNN Layer × L) | (4096, 128) |
| | BatchNorm + ReLU | (4096, 128) |
| Memory bank | Construction | $(M, 128)$ |
| | Linear + LayerNorm | (4096, 128) |

Table 18: Details of SPARK's downstream architecture.

| Downstream | | |
|---|---|---|
| **Procedure** | **Layer** | **Dimension** |
| Augmentation | GNN Encoder | $(T_0, 4096, 128)$ |
| | Memory bank retrieval | $(T_0, 4096, 128)$ |
| Historical observations encoding | Attention score of time steps | $(, T_0)$ |
| | Initial state encoding | (1, 4096, 128) |
| Fourier-enhanced graph ODE | Fourier transform | (1, 4096, 128) |
| | Linear transform | (1, 4096, 128) |
| | Inverse Fourier transform | (1, 4096, 128) |
| | ODE solver | $(T, 4096, 128)$ |

Table 19: Detailed setup for OOD experiments.

| Equation/Model | Training Parameters | Testing Parameters |
|---|---|---|
| 2D Navier-Stokes Equation | $\nu = \{1e^{-1}, 1e^{-2}, \ldots, 1e^{-7}, 1e^{-8}\}$ | $\nu = \{1e^{-9}, 1e^{-10}\}$ |
| Spherical Shallow Water Equation | $\nu = \{1e^{-1}, 1e^{-2}, \ldots, 1e^{-7}, 1e^{-8}\}$ | $\nu^t = \{1e^{-9}, 1e^{-10}\}$ |
| 3D Reaction-Diffusion Equations | $D = \{2.1 \times 10^{-5}, 1.6 \times 10^{-5}, 6.1 \times 10^{-5}\}$ | $D = \{2.03 \times 10^{-9}, 1.96 \times 10^{-9}\}$ |
| ERA5 | $V = \{Sp, SST, SSH, T2m\}$ | $V = \{SSR, SSS\}$ |

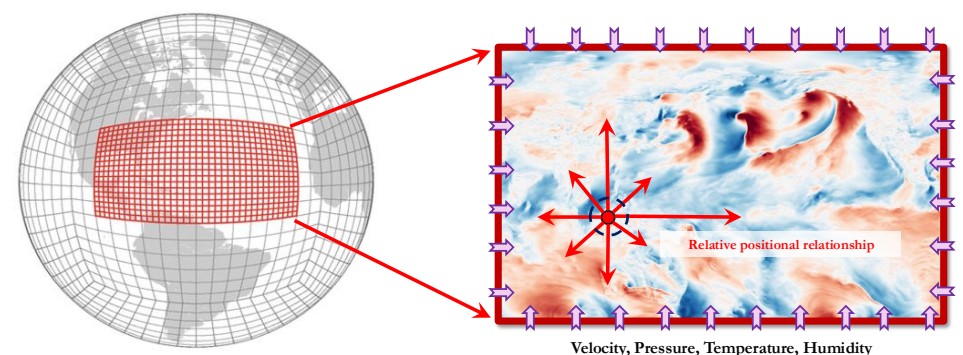

Figure 8: Example of Boundary Information on ERA5.

