# OpenReview forum: "SPARK: Physics-Guided Quantitative Augmentation for Dynamical System Modeling"
_ICLR.cc/2025/Conference — Submitted to ICLR 2025_

### Official Review · Reviewer_pH3K · 2024-11-01

**Soundness:** 3
**Presentation:** 3
**Contribution:** 3
**Rating:** 6
**Confidence:** 4

**Summary:**

This paper proposes SPARK to address the challenges of data scarcity and distribution shifts in dynamical system modeling. SPARK integrates boundary information and physical parameters by using an autoencoder, and then a pre-trained memory bank is obtained. It further combines Fourier-enhanced graph ODE to efficiently predict long-term dynamical systems. The experimental results have demonstrated the superiority of the proposed method against the baseline models across many dynamical systems under distribution shifts and limited data conditions.

**Strengths:**

- This paper presents an interesting idea for handling data scarcity and OOD, which are important topics in scientific machine learning.

- This paper is well-written and has a detailed presentation of methods, experimental setup, and results discussion.

- This paper has tested multiple challenging datasets, such as ERA5 and 3D systems.

**Weaknesses:**

- The motivation for using each component in SPARK can be further clarified. The paper will benefit from discussing the interconnection between each network component.

- It would also be good to have ablation studies on incorporated physics. The authors may consider reducing physical information (i.e., boundary information and physical parameters) for pre-training. Then, we can see the contribution of each physical component.

**Questions:**

- On Page 6, for RQ2, could you be more specific on what challenging tasks?

- What is the setup for OOD experiments?

- How do you compute PSNR and SSIM for scientific data? Image data has a fixed range of [0,255] but scientific data doesn’t.

- Energy Spectrum is a common metric for fluid dynamics. Is it also commonly used for reaction-diffusion equations? How does this paper compute the energy spectrum?


- Some minor typos:

    - On Page 2, “effectively long-term prediction” should be “effective …”.

---

> ### Author Response · Authors · 2024-11-21
> **Response to Reviewer pH3K (Part 1/3)**
>
> Dear Reviewer pH3K,
>
> We sincerely appreciate the time you’ve dedicated to reviewing our paper, as well as your valuable insights and support. Your positive feedback is highly motivating for us. Below, we address your primary concern and offer further clarification.
>
> ---
>
> > **Q1**. The motivation for using each component in SPARK can be further clarified. The paper will benefit from discussing the interconnection between each network component.
>
> **A1**. Thank you for your comment. In this work, we design a unified plugin SPARK with three basic modules:
>
> - **Physics-incorporated data compression**. We integrate physical parameters, boundary information and input observations into latent space through position encoding and channel attention.
>
> - **Memory bank construction**. We then pre-train a discrete memory bank through GNN-based reconstruction and discrete quantization mechanism.
>
> - **Downstream augmentation and prediction**. We froze the memory bank's weights and use it to realize augmentation, which introduces diversity into the training set. Further, we design a fourier-enhanced graph ODE to accurately predict complex dynamics.
>
> These three designs contribute jointly to the high accuracy and strong generalization ability in various environments. We will add disscussions about components' interconnection in the revised manuscript.
>
> > **Q2**. It would also be good to have ablation studies on incorporated physics. The authors may consider reducing physical information (i.e., boundary information and physical parameters) for pre-training. Then, we can see the contribution of each physical component.
>
> **A2**. Thank you for your valuable feedback. We add ablation experiments on incorporated physics by reducing different physical information. The experiments are conducted on OOD scenarios, and the results are shown below. The results demonstrate the effectiveness of each physical component. We will include it in our revised manuscript.
>
> |  | Ours | w/o parameter | w/o boundary | w/o parameter&boundary |
> | ------------- | ---------- | ------ | ------ | ------ |
> | PROMETHEUS    | **0.0301** | 0.0357 | 0.0324 | 0.0397 |
> | NAVIER–STOKES | **0.0725** | 0.0833 | 0.0764 | 0.0902 |

---

> > ### Author Response · Authors · 2024-11-21
> > **Response to Reviewer pH3K (Part 2/3)**
> >
> > > **Q3**. On Page 6, for RQ2, could you be more specific on what challenging tasks?
> >
> > **A3**. Thank you for your valuable feedback. We first redefine challenging tasks in dynamical modeling prediction as problems that arise due to the inherent complexities of capturing high-dimensional, nonlinear, and chaotic systems[1,2]. These tasks often require models to adapt across real-world scenarios, like extreme events or long-term prediction.
> >
> > In section 4.3, we focus on the prediction of sea ice evolution. This is challenging due to the complex, nonlinear interactions governing its Lagrangian motion[3], coupled with the spatiotemporal variability of environmental forcing factors.
> >
> > To dispel your concerns, referencing[4], we add two challenging experiments, namely long-term prediction and extreme event prediction. We choose Prometheus and Sevir datasets to conduct the two experiments, separately.
> >
> > - **For long-term prediction**, we use Prometheus with ten steps as input and supervise the prediction of the next ten steps during training. During inference, we predict the next 10, 30, and 50 steps in an autoregressive manner. The table below demonstrates that our model outperforms other baselines in long-term prediction performance.
> >
> > |  Time step  | U-Net  | ViT    | FNO    | NMO    | Ours       |
> > | ----------- | ------ | ------ | ------ | ------ | ---------- |
> > |    10       | 0.0931 | 0.0674 | 0.0447 | 0.0397 | **0.0294** |
> > |    30       | 0.1374 | 0.1038 | 0.0815 | 0.0726 | **0.0537** |
> > |    50       | 0.2238 | 0.1842 | 0.1374 | 0.1154 | **0.0921** |
> >
> > - **For extreme event prediction**, we use Sevir dataset, which contains data related to severe weather phenomena. To better evaluate the prediction performance of extreme events, we used the Critical Success Index (CSI) as in[2], in addition to MSE. For simplicity, we used only the thresholds {16, 133, 181, 219} and the mean CSI-M. The table below shows that our model consistently outperform these baselines in extreme event prediction. We will include these in our revised version.
> >
> > | Model | CSI-M $\uparrow$ | CSI-219 $\uparrow$ | CSI-181 $\uparrow$ | CSI-133 $\uparrow$ | CSI-16 $\uparrow$ | MSE($10^{-3}$)$\downarrow$ |
> > | ------- | ------ | ------ | ------ | ------ | ------ | ------ |
> > | U-Net   | 0.3593 | 0.0577 | 0.1580 | 0.3274 | 0.7441 | 4.1119 |
> > | ViT     | 0.3692 | 0.0965 | 0.1892 | 0.3465 | 0.7326 | 4.1661 |
> > | PredRNN | 0.4028 | 0.1274 | 0.2324 | 0.3858 | 0.7507 | 3.9014 |
> > | SimVP   | 0.4275 | 0.1492 | 0.2538 | 0.4084 | 0.7566 | 3.8182 |
> > | Ours    | **0.4683** | **0.1721** | **0.2734** | **0.4375** | **0.7792** | **3.6537** |
> >
> >
> > > **Q4**. What is the setup for OOD experiments?
> >
> > **A4**. Thank you for your comment. We propose that training and testing in the in-domain parameters is called w/o OOD experiments, while training in the in-domain parameters and testing in the out-domain parameters is called w/ OOD experiments. Here we present the in-domain and out-domain parameters for different benchmarks in the table below. We will include it in our revised manuscript.
> >
> > | Benchmarks | In-Domain Parameters   | Out-Domain Parameters   |
> > |------------|------------------------|-------------------------|
> > | PROMETHEUS | $(a_1, a_2, \ldots, a_{25})$, $(b_1, b_2, \ldots, b_{20})$ | $(a_{26}, a_{27}, \ldots, a_{30})$, $(b_{21}, b_{22}, \ldots, b_{25}\)$ |
> > | 2D Navier-Stokes Equation       | $ν = (1e^{-1}, 1e^{-2}, \ldots, 1e^{-7}, 1e^{-8}) $ | $ν = (1e^{-9}, 1e^{-10})$ |
> > | Spherical Shallow Water Equation  | $ν = (1e^{-1}, 1e^{-2}, \ldots, 1e^{-7}, 1e^{-8}) $   | $ν = (1e^{-9}, 1e^{-10}) $   |
> > | 3D Reaction-Diffusion Equations   | $D = (2.1 × 10^{-5}, 1.6 × 10^{-5}, 6.1 × 10^{-5})$ | $D = (2.03 × 10^{-9}, 1.96 × 10^{-9}) $ |
> > | ERA5  | $V = ({Sp, SST, SSH, T2m})$  | $V = ({SSR, SSS})$ |
> >
> >
> > ---
> > [1] Wu H, et al. "Solving high-dimensional pdes with latent spectral models." ICML2023.
> >
> > [2] Gao Z, et al. "Earthformer: Exploring space-time transformers for earth system forecasting." NeurIPS2022.
> >
> > [3] Notz D. "Challenges in simulating sea ice in Earth System Models." Wiley Interdisciplinary Reviews: Climate Change, 2012.
> >
> > [4] Wang K, et al. "NuwaDynamics: Discovering and Updating in Causal Spatio-Temporal Modeling." ICLR2024.

---

> > > ### Author Response · Authors · 2024-11-21
> > > **Response to Reviewer pH3K (Part 3/3)**
> > >
> > > > **Q5**. How do you compute PSNR and SSIM for scientific data? Image data has a fixed range of \[0,255\] but scientific data doesn’t.
> > >
> > > **A5**. Thank you for your comment. We adapt the calculation of PSNR and SSIM for scientific data, which does not have a fixed range like image data. By normalizing the data based on its dynamic range in each experiment, we ensure the calculations align with traditional definitions and remain comparable across different datasets. This method preserves the physical meaning of the values and provides accurate quantitative assessments of prediction quality.
> > >
> > > > **Q6**. Energy Spectrum is a common metric for fluid dynamics. Is it also commonly used for reaction-diffusion equations? How does this paper compute the energy spectrum?
> > >
> > > **A6**. Thank you for your insightful feedback. Energy spectrum analysis, which is widely used in fluid dynamics to characterize energy distribution across spatial scales, is equally applicable to 3D Reaction-Diffusion Equations. The 3D Reaction-Diffusion Equations model diffusion and reaction processes in space using partial differential equations. To compute the energy spectrum, we apply Fourier transforms to decompose spatial variables into wave number components in the frequency domain. Specifically, we calculate the energy spectrum using the formula:
> > >
> > > $E(k) = \sum\_{|\mathbf{k}| = k} \frac{1}{2} |\hat{u}(\mathbf{k})|^2, \quad \hat{u}(\mathbf{k}) = \int u(\mathbf{x}) e^{-i \mathbf{k} \cdot \mathbf{x}} \, d\mathbf{x},$
> > >
> > > where $\mathbf{k}$ denotes the wave vector, and $|\mathbf{k}|$ corresponds to the wave number. This approach ensures a robust quantitative analysis of spatial energy distributions.
> > >
> > > > **Q7**. Some minor typos: On Page 2, “effectively long-term prediction” should be “effective …”.
> > >
> > > **A7**. Thank you for your comment. We have thoroughly reviewed the entire manuscript, and have corrected the mentioned or other minor errors to enhance the paper's clarity and precision.
> > >
> > > ---
> > > Thanks again for appreciating our work and for your constructive suggestions. Please let us know if you have further questions.
> > >
> > > Best,
> > >
> > > the Authors

---

> > > > ### Author Response · Authors · 2024-11-24
> > > > **Kindly Request for Feedback of Reviewer**
> > > >
> > > > Dear Reviewer pH3K,
> > > >
> > > > As the rebuttal deadline is coming soon, please let us know if our responses have addressed your main concerns. If so, we kindly ask for your reconsideration of the score. If any aspects require additional elaboration or refinement, we will be more than happy to engage in further improvements and discussion.
> > > >
> > > > Thanks again for your time.

---

> > > > > ### Author Response · Authors · 2024-11-25
> > > > > **Thank you & Looking forward to further discussion**
> > > > >
> > > > > Dear Reviewer pH3K,
> > > > >
> > > > > We deeply appreciate your dedication to engaging in author-reviewer discussions. To facilitate better understanding of our rebuttal and revision, we have outlined your key concerns and our responses to enhance communication:
> > > > >
> > > > > - **About the motivation for using each component in SPARK.**
> > > > > We have included discussion about the interconnection between each network component and conducted ablation experiments to demonstrate the contribution of each physical component.
> > > > >
> > > > > - **About the challenging tasks.**
> > > > > We have redefined challenging tasks in dynamical system modeling and explained why the prediction of sea ice is challenging. Further, we choose two specific challenging tasks (long-term prediction and extreme event prediction) and conduct experiments to demonstrate the effectiveness of SPARK.
> > > > >
> > > > > - **About OOD experimental setup.**
> > > > > We have provided more detailed OOD experimental setup descriptions and the corresponding table.
> > > > >
> > > > > For other issues not mentioned here, please refer to our detailed rebuttal response. We sincerely hope this addresses your concerns! We humbly look forward to further discussion with you.
> > > > >
> > > > > Thank you again for your valuable guidance and thoughtful review.
> > > > >
> > > > > Warm regards,
> > > > >
> > > > > Authors

---

> > > > > > ### Author Response · Authors · 2024-11-28
> > > > > > **Thank you & Looking forward to further discussion!**
> > > > > >
> > > > > > Dear Reviewer pH3K,
> > > > > >
> > > > > > We sincerely thank you for your valuable and constructive feedback! Since the Discussion Period Extension provides us with additional time, we are eager to address any further concerns you may have. If our current response satisfactorily resolves your main concerns, we kindly ask for your reconsideration of the score. Should you have any further advice on the revised paper and/or our rebuttal, please let us know, and we will be more than happy to engage in further discussion and improve the paper.
> > > > > >
> > > > > > Thank you so much for devoting time to improving our paper!
> > > > > >
> > > > > > Best regards,
> > > > > >
> > > > > > the Authors

---

> > > > > > > ### Comment · Reviewer_pH3K · 2024-11-28
> > > > > > >
> > > > > > > Thanks for your rebuttal. My concerns have been addressed. I will maintain my score of 6.

---

> > > > > > > > ### Author Response · Authors · 2024-12-02
> > > > > > > > **Thank You for Your Feedback and Support**
> > > > > > > >
> > > > > > > > Dear Reviewer pH3K,
> > > > > > > >
> > > > > > > > Thank you for your insightful review and dedication to the review process. We are glad to see that all your concerns have been addressed! We remain eager to address any further questions or concerns you may have.
> > > > > > > >
> > > > > > > > Warm regards,
> > > > > > > >
> > > > > > > > Authors

---

### Official Review · Reviewer_bjTD · 2024-11-02

**Soundness:** 3
**Presentation:** 2
**Contribution:** 2
**Rating:** 3
**Confidence:** 4

**Summary:**

The paper introduces SPARK, a physics-guided augmentation framework for modeling dynamical systems that overcomes the limitations of traditional numerical and data-driven methods. By incorporating a unique compression and augmentation plugin, along with an attention mechanism and Fourier-enhanced graph ODE, SPARK improves model generalization and robustness, especially in data-scarce situations and distribution shifts. Experimental results highlight SPARK's strong performance in accurately predicting complex spatiotemporal dynamics, particularly in challenging cases like sea ice evolution, effectively capturing intricate physical phenomena.

**Strengths:**

1. By incorporating boundary information and physical parameters, SPARK enhances the model's ability to generalize across different physical scenarios, which is crucial for real-world applications.
2. The paper provides extensive experimental results across various benchmark datasets, demonstrating SPARK's superior performance compared to existing models, particularly in handling out-of-distribution scenarios.

**Weaknesses:**

1. The symbols and formulas appear to be somewhat disorganized, which makes it difficult for readers to understand the meaning.  Clear definitions and a more structured presentation of the equations would greatly enhance the paper's accessibility and overall readability.
2. The lack of novelty. This paper claims to be the first to use physics-guided compression and augmentation. But there has been a paper [1] doing like this. The techniques of the two papers are very similar, including : (1) using VQ-VAE to compress information (2) augmenting training set by the top-K discrete embeddings.
3. The proposed methodology, may be complex to implement in practice. The paper could provide more guidance or examples on how to effectively apply SPARK in different contexts.

[1] Wu, Hao, et al. "BeamVQ: Aligning Space-Time Forecasting Model via Self-training on Physics-aware Metrics." arXiv preprint arXiv:2405.17051 (2024).

**Questions:**

1. In line 163, it needs references for those methods which simply concatenate boundary information with node features.
2. What does boundary information refer to? Give some examples please.
3. In abstract, what's the meaning of "stable data distribution"? Provide explanations about it and why does it can cause ineffectiveness of data scarcity and distribution shifts.

---

> ### Author Response · Authors · 2024-11-21
> **Response to Reviewer bjTD (Part 1/4)**
>
> Dear Reviewer bjTD,
>
> We sincerely appreciate the time you’ve dedicated to reviewing our paper, as well as your valuable insights and support. Below, we address your primary concern and offer further clarification.
>
> ---
> > **Q1**. The symbols and formulas appear to be somewhat disorganized, which makes it difficult for readers to understand the meaning. Clear definitions and a more structured presentation of the equations would greatly enhance the paper's accessibility and overall readability.
>
> **A1**. Thank you for your valuable feedback. To better facilitate the understanding of our paper, we have made the following modifications:
>
> - **Problem definition.** We have refined the problem definition and ensured consistency throughout the manuscript. As follows:
>
> "Given a dynamical system governed by physical laws such as PDEs, we aim to enhance prediction using autoencoder reconstruction and discrete quantization. We have $N$ observation points in the domain $\Omega$, located at $\mathbf{s} = \{\mathbf{s}\_1, \cdots, \mathbf{s}\_N\}$, where $\mathbf{s}\_i \in \mathbb{R}^{d\_s}$. At time step $t$, the observations are $\mathcal{X}^t = \{\mathcal{X}\_1^t, \cdots, \mathcal{X}\_N^t\}$, where $\mathcal{X}\_i^t \in \mathbb{R}^{d}$ and $d$ represents the number of observation channels. Boundary information and physical parameters affect the dynamical system, leading to different conditions and distribution shifts. We first employ reconstruction model and construct a discrete memory bank to compress and store physical prior information. Then, given historical observation sequences {$\{\mathcal{X}\_i^{-T\_0+1:0}\}$}$\_{i=1}^N$, our goal is to use the pre-trained memory bank for data augmentation and predict future observations {$\{\mathcal{Y}\_i^{1:T}\}$}$\_{i=1}^N$ at each observation point."
>
> - **Symbols and formulas.** We thoroughly review all symbols and formulas in the manuscript to ensure their meanings are precise and clear. For instance, we make the following modifications:
>
> $\boldsymbol{u}_i = \text{Proj} \left( \mathcal{X}_i , \boldsymbol{p}^{rel}_i \right) \quad \text{with} \quad
>     \boldsymbol{p}^{rel}_i = \phi\left( \mathbf{s}_i, \boldsymbol{p}^{boun}_i \right), \quad (1)$
>
> $\mathcal{L}\_{pre}=\frac{1}{T N} \sum\_{t=1}^T \sum\_{i=1}^N\left(\hat{\mathcal{X}}\_{i}^{t}-\mathcal{X}\_{i}^{t}\right)^2+ \frac{1}{T N} \sum\_{t=1}^T \sum\_{i=1}^N\left (\mu \left\|\boldsymbol{h}\_{i}^{t}-\mathbf{s g}[\boldsymbol{e}]\right\|\_2^2+\gamma\left\|\mathbf{s g}\left[\boldsymbol{h}\_{i}^{t}\right]-\boldsymbol{e}\right\|\_2^2\right ), \quad (6)$
>
> $\boldsymbol{q}\_{i}=\frac1{T\_0}\sum\_{t=1}^{T\_0}\delta(\alpha\_{i}^{t}
> \cdot \boldsymbol{v}\_{i}^{t}),\quad \alpha\_{i}^{t}= \left(\boldsymbol{v}\_{i}^{t} \right )^{T} \cdot \mathrm{tanh}\left(\left(\frac{1}{T\_0} \sum\_{t=1}^{T\_0} \boldsymbol{v}\_{i}^{t}\right)W\_{\alpha}\right), \quad (8)$
>
> $\mathcal{L}\_{\text{dyn}} = \frac{1}{TN} \sum\_{i=1}^T \sum\_{i=1}^N \|\hat{\mathcal{Y}}\_{i}^{t} - \mathcal{Y}\_{i}^{t}\|\_2^2 + \lambda\_{\text{reg}} \mathcal{R}(\theta). \quad (11)$

---

> > ### Author Response · Authors · 2024-11-21
> > **Response to Reviewer bjTD (Part 2/4)**
> >
> > > **Q2**. Problem about novelty.
> >
> > **A2**. Thank you for your valuable comments. To the best of our knowledge, SPARK should be the first physics-guided compression and augmentation framework. There are fundamental differences between SPARK and BeamVQ[1]:
> >
> > **(1) Difference in compressing data with VQ-VAE.**
> >
> > - > **Different input.** SPARK's input includes physical prior information (boundary information and physical parameters) along with the input data, while BeamVQ's input consists only of the observational data.
> >
> > - > **Different workflows and SPARK is more lightweight.** While both SPARK and BeamVQ are plugin-like frameworks, upon carefully reviewing the original paper, we find that BeamVQ actually functions as an end-to-end framework. In BeamVQ, the encoder acts as the backbone model. This design introduces a large number of parameters during training, which aligns with the original authors' claim of requiring 16 NVIDIA A100-PCIE-40GB GPUs. Essentially, BeamVQ can be considered a combination of a backbone and an improved VQVAE, making it not strictly a plugin. This likely affects the original backbone. In contrast, SPARK is a two-stage framework where the memory bank, once trained, is frozen and directly used for downstream tasks. This is quite lightweight. We have included a schematic comparison of SPARK and BeamVQ in the appendix to clearly illustrate their differences.
> >
> > - > **Experiment comparison.** To address your concerns, we contact the authors of the BeamVQ paper and obtain partial access to their codes. The core codes of BeamVQ is provided in Appendix H.9. We then conduct experiments on the Navier–Stokes, Spherical-SWE, Prometheus, and 3D Reaction–Diff dataset. Here, we use FNO and SimVP as backbones. Further, we select parameter count, training time, and inference time to compare the effiency of the two models on Navier–Stokes. The table below shows that SPARK is much more lightweight and performs better, supporting our claims. Notably, the SimVP+BeamVQ model variant crashes on 3D Reaction-Diff due to memory overflow, as its parameter complexity is unsuitable for 3D scenarios. We will include these in our revised version.
> >
> > |              |Navier–Stokes|Spherical-SWE|Prometheus| 3D Reaction–Diff |
> > | ------------ | ------ | ------ | ------ | ------ |
> > | FNO          | 0.1556 | 0.0038 | 0.0447 | 0.0132 |
> > | FNO+BeamVQ   | 0.1342 | 0.0032 | 0.0356 | 0.0104 |
> > | FNO+SPARK    | 0.1257 | 0.0029 | 0.0338 | 0.0095 |
> > | SimVP        | 0.1262 | 0.0031 | 0.0394 | 0.0108 |
> > | SimVP+BeamVQ | 0.1173 | 0.0027 | 0.0375 | -      |
> > | SimVP+SPARK  | 0.1105 | 0.0024 | 0.0360 | 0.0087 |
> >
> > |   |MSE|Param| Training time | Inference time |
> > | ------------ | ------ | ------ | ------ | ------ |
> > | FNO+BeamVQ   | 0.1342 | 214.25 MB | 26.11 h | 3.25 s  |
> > | FNO+SPARK    | 0.1257 | 35.67 MB  | 4.2 h   | 0.58 s |
> >
> > **(2) Difference in augmenting data by the top-K discrete embeddings.**
> >
> > - > **Different usage of top-K.** BeamVQ relies on high-quality, non-differentiable physical metrics for filtering, which are not available in all scenarios and require domain-specific expertise. In contrast, SPARK's top-k approach aims to expand the search space, and we use the fusion of input with the top-k embeddings.
> >
> > - > **Experiments on hyperparameter $k$.** To address your concerns, we add experiments on the value of $k$ on the Navier-Stokes，Prometheus, 3D Reaction–Diff, and ERA5 datasets. The candidate values are \{1,3,5,7,9,11\}, and the results are shown in the table below.
> >
> > |              |Navier–Stokes|Spherical-SWE|Prometheus| 3D Reaction–Diff |
> > | ------------ | ------ | ------ | ------ | ------ |
> > | k=1          | 0.0752 | 0.0022 | 0.0315 | 0.0116 |
> > | k=3          | 0.0726 | **0.0018** | **0.0296** | 0.0108 |
> > | k=5          | **0.0715** | 0.0021 | 0.0303 | **0.0104** |
> > | k=7          | 0.0731 | 0.0024 | 0.0311 | 0.0110 |
> > | k=9          | 0.0764 | 0.0025 | 0.0320 | 0.0121 |
> > | k=11         | 0.0780 | 0.0029 | 0.0327 | 0.0128 |
> >
> > As $k$ increases, the model's performance first improves and then declines, with optimal performance generally achieved when $k$ is between 3 and 5.
> >
> > In summary, our method is a genuine two-stage framework. Compared to BeamVQ, SPARK is more lightweight and achieves better performance. The complete results will be included in the revised version.
> >
> > ---
> > [1] Wu H, et al. "BeamVQ: Aligning Space-Time Forecasting Model via Self-training on Physics-aware Metrics." arXiv.

---

> > > ### Author Response · Authors · 2024-11-21
> > > **Response to Reviewer bjTD (Part 3/4)**
> > >
> > > > **Q3**. The proposed methodology, may be complex to implement in practice. The paper could provide more guidance or examples on how to effectively apply SPARK in different contexts.
> > >
> > > **A3**. Thank you for your comment. Our framework uses a two-stage design. We demonstrate its application with experiments on the Navier–Stokes equations.
> > >
> > > - **In the pretraining stage**, the model takes physical parameters and boundary information as inputs. Once training converges, we save the memory bank for retrieval and augmentation in downstream tasks.
> > >
> > > - **In the downstream stage**, we use different backbones for spatiotemporal prediction. Our experiments include FNO, CNO, and SimVP as backbones, and the results are shown in the table.
> > >
> > > | Backbone      | MSE      | SSIM     |
> > > |---------------|----------|----------|
> > > | FNO           | 0.1556   | 0.923    |
> > > | FNO + SPARK   | 0.1257   | 0.936    |
> > > | CNO           | 0.1473   | 0.938    |
> > > | CNO + SPARK   | 0.1341   | 0.945    |
> > > | SimVP         | 0.1262   | 0.957    |
> > > | SimVP + SPARK | 0.1105   | 0.962    |
> > >
> > > - **Flexible plugin.** The method supports transfer learning. For example, the memory bank pretrained on the Navier–Stokes equations transfers directly to the Spherical-SWE equations for retrieval and augmentation. The results are shown in the table below.
> > >
> > >
> > > | Task       | Memory Bank Source      | MSE    | SSIM  |
> > > |   -        |           -             | -      |  -    |
> > > | Spherical-SWE | Navier–Stokes Equations | 0.0027 | 0.948 |
> > >
> > >
> > > In summary, the method is lightweight and works as a plugin that integrates seamlessly with any baseline prediction model. To enhance your understanding, we summarize the detailed processing steps in the table below.
> > >
> > > | Stage       | Description                                         | Inputs                        | Target           |
> > > |-------------|-----------------------------------------------------|-------------------------------|-------------------|
> > > | Pretraining | Train model with physical parameters and boundaries | Physical parameters, Boundary information | Memory Bank       |
> > > | Downstream  | Spatiotemporal prediction backbone | Memory Bank, Input data       | Prediction Results|
> > > | Transfer Learning | Transfer Memory Bank to new task for augmentation and prediction | Pretrained Memory Bank, New task data | Enhanced Predictions|
> > >
> > > > **Q4**. In line 163, it needs references for those methods which simply concatenate boundary information with node features.
> > >
> > > **A4**. Thank you for your comment. We have carefully reviewed the relevant literature and have included appropriate references[1,2] in the revised manuscript to support this.
> > >
> > > ---
> > > [1] Wang H, et al. "BENO: Boundary-embedded Neural Operators for Elliptic PDEs." ICLR2024.
> > >
> > > [2] Lötzsch W, et al. "Learning the solution operator of boundary value problems using graph neural networks." ICML2022.

---

> > > > ### Author Response · Authors · 2024-11-21
> > > > **Response to Reviewer bjTD (Part 4/4)**
> > > >
> > > > > **Q5**. What does boundary information refer to? Give some examples please.
> > > >
> > > > **A5**. Thank you for your comment. In dynamical systems and natural sciences, boundary information is mathematical term used to describe the behavior of physical systems at their boundaries.
> > > >
> > > > In our paper, boundary information is divided into geometric boundary position information and intrinsic features at the corresponding boundary. The geometric boundary position refers to the **relative distance** between the **current node** and **the nearest boundary point**. And the intrinsic features at the boundary vary with the system. For example, in the ERA5 dataset, these features include velocity, pressure, temperature, and humidity. To facilitate understanding, we use ERA5 dataset as an example and visualize the boundary information in Appendix I.
> > > >
> > > >
> > > > > **Q6**. In abstract, what's the meaning of "stable data distribution"? Provide explanations about it and why does it can cause ineffectiveness of data scarcity and distribution shifts.
> > > >
> > > > **A6**. Thank you for your insightful comment. By "stable data distribution", we refer to the assumption that the training and testing samples are drawn from the same probability distribution, i.e., $P\_{\text{train}}(X, Y) = P\_{\text{test}}(X, Y).$
> > > >
> > > > Under this assumption, minimizing the empirical risk on the training set leads to good generalization on the test set:
> > > >
> > > > $R_{\text{emp}}(f) = \frac{1}{n} \sum\_{i=1}^{n} L(f(X\_i), Y\_i),$
> > > > $R(f) = \mathbb{E}\_{(X, Y) \sim P\_{\text{test}}(X, Y)} [L(f(X), Y)],$
> > > >
> > > > where $L(\cdot, \cdot)$ is the loss function and $f$ is the learned model. However, **data scarcity** makes it difficult to accurately estimate $P\_{\text{train}}(X, Y)$ because the sample size $n$ is too small. This increases the discrepancy between the empirical distribution $\hat{P}\_{\text{train}}(X, Y)$ and the true distribution $P\_{\text{train}}(X, Y)$, leading to higher generalization error.
> > > >
> > > > Furthermore, **distribution shift** directly cause, i.e., $P\_{\text{train}}(X, Y) \neq P\_{\text{test}}(X, Y).$ As a result, the model optimized on the training data performs poorly on the testing data.
> > > >
> > > >
> > > > ---
> > > > Thanks again for your constructive suggestions! Please let us know if you have further questions.
> > > >
> > > > Best,
> > > >
> > > > the Authors

---

> > > > > ### Author Response · Authors · 2024-11-28
> > > > > **Thank you & Looking forward to further discussion!**
> > > > >
> > > > > Dear Reviewer bjTD,
> > > > >
> > > > > We sincerely thank you for your valuable and constructive feedback! Since the Discussion Period Extension provides us with additional time, we are eager to address any further concerns you may have. If our current response satisfactorily resolves your main concerns, we kindly ask for your reconsideration of the score. Should you have any further advice on the revised paper and/or our rebuttal, please let us know, and we will be more than happy to engage in further discussion and improve the paper.
> > > > >
> > > > > Thank you so much for devoting time to improving our paper!
> > > > >
> > > > > Best regards,
> > > > >
> > > > > the Authors

---

> > > > > > ### Author Response · Authors · 2024-12-01
> > > > > > **Respectful Inquiry Before Discussion Deadline**
> > > > > >
> > > > > > Dear reviewer bjTD,
> > > > > >
> > > > > > Thank you for taking the time and effort to provide a valuable review of our work. As we are approaching the end of the discussion, we hope that you have had the chance to review our previous response. If our response has addressed your concerns, we thank you for reconsidering the score, and we are more than willing to engage in further discussion if needed.
> > > > > >
> > > > > > Yours sincerely,
> > > > > >
> > > > > > Authors

---

> > > > > > > ### Author Response · Authors · 2024-12-02
> > > > > > > **[ Only 1 Day Remaining ] A Gentle Reminder of Feedbacks**
> > > > > > >
> > > > > > > Dear Reviewer bjTD,
> > > > > > >
> > > > > > > We sincerely apologize for reaching out again and fully understand that your time is extremely valuable. With the discussion deadline so close, we are eager to know if our responses have alleviated your concerns.
> > > > > > >
> > > > > > > We are pleased to see that Reviewer 89BR has increased the score, and we are glad to have addressed all his concerns. We appreciate the recognition of our work by other three reviewers, and their positive feedbacks like "an intelligent design choice" (Reviewer Rucb), "well-written and well-presented" (Reviewer 89BR), "an interesting idea" (Reviewer pH3K).
> > > > > > >
> > > > > > > In our previous response, we have provided detailed answers to your concerns, including: (1) an explanation of the differences between our paper and the one you mentioned, along with corresponding comparative experiments; (2) a clear definition of boundary information; and (3) an explanation of what constitutes a stable data distribution.
> > > > > > >
> > > > > > > To facilitate understanding, we would like to clarify the contributions of our paper once again. We propose a reconstruction-based vector quantization technique to compress rich boundary information and physical parameters, which we then leverage for physics-guided data augmentation. In downstream task, we incorporate an attention mechanism to model historical observations and design a Fourier-enhanced graph ODE for precise and efficient forecasting. Our work aims to contribute to addressing the challenges of data scarcity and out-of-distribution generalization in the modeling of dynamical systems.
> > > > > > >
> > > > > > > We have carefully refined the manuscript following your insightful feedbacks. Lastly, we would be most grateful if you could kindly reconsider your rating!
> > > > > > >
> > > > > > > Thank you again for your invaluable guidance and thoughtful review.
> > > > > > >
> > > > > > > Warm regards,
> > > > > > >
> > > > > > > Authors

---

> > > > > > > > ### Comment · Reviewer_bjTD · 2024-12-03
> > > > > > > >
> > > > > > > > Thanks for the rebuttal. I will take these into consideration.

---

> > > > > > > > > ### Author Response · Authors · 2024-12-03
> > > > > > > > >
> > > > > > > > > Dear Reviewer bjTD,
> > > > > > > > >
> > > > > > > > > Thank you once again for your response. Your feedback is incredibly valuable to us. We sincerely request you to reconsider your evaluation and extend our heartfelt gratitude for your time and effort.
> > > > > > > > >
> > > > > > > > > Thank you again for your invaluable guidance and thoughtful review.
> > > > > > > > >
> > > > > > > > > Warm regards,
> > > > > > > > >
> > > > > > > > > Authors

---

### Official Review · Reviewer_89BR · 2024-11-03

**Soundness:** 3
**Presentation:** 2
**Contribution:** 3
**Rating:** 6
**Confidence:** 4

**Summary:**

Data-driven methods for dynamical systems often face distribution shift challenges. To tackle this, this paper proposes SPARK, a physics-guided plugin to address both environmental distribution shift (due to changes in boundary conditions and physical parameters) and temporal distribution shift.
SPARK achieves this by incorporating the boundary information and physical parameters into a discrete memory bank constructed through solution reconstructions.
By embedding these physical priors, the memory bank can then be used to augment data samples in downstream tasks, thereby increasing model generalizability.
To handle the temporal distribution shift, SPARK encodes historical information into initial states through attention and uses Fourier-enhanced graph ODE for long-term prediction.
In the end, the paper evaluates SPARK on several benchmarks.

**Strengths:**

1. The idea of increasing model generalizability by augmenting data in downstream tasks with a pre-trained memory bank that contains boundary information and physical parameters is nice.
2. The paper evaluates the method on a good number of benchmarks.

**Weaknesses:**

1. The degree of originality is not high. It shares quite some similarities with the DGODE model in (Prometheus by Wu et al. 2024, cited by the paper), which proposed the idea of "codebank" to include the environmental factor for OOD and graph ODE for future predictions.
2. The algorithm is not clearly presented. The paper presented reasonable ideas but without enough technical details to tell a clear story.
Mathematical notation is not clearly defined, which makes it hard to follow the method. For example, how is the "real boundary" ("p^{boun}")represented? Is it a list of spatial coordinates of discrete boundary nodes? Time index does not make sense in section 3.3. For example, T in equation 8 is the length of history observations but then it is also used in loss function in equation 11 to represent the number of future predictive steps, which is confusing. Index notation does not make sense in the pretraining loss equation (6).
3. It's unclear how the discrete memory bank is built. What are the e_i in E and how are they constructed ?

**Questions:**

1. How are the with and without OOD datasets constructed in the experiments? Is there an explanation for why SPARK achieved better performance on OOD cases even than other models did on non-OOD cases?
2. Given the big accuracy difference, what is the training cost comparison?
3. What are the training details, e.g., model architecture, optimizer, training devices?  Is boundary information injected at two places, i.e., through node features and directly through the boundary latent vector B?

---

> ### Author Response · Authors · 2024-11-21
> **Response to Reviewer 89BR (Part 1/3)**
>
> Dear Reviewer 89BR,
>
> Thank you for your valuable feedback on our manuscript! We have taken your comments seriously and have made the necessary revisions and additions to address the concerns raised.
>
> ---
> > **Q1**. Problem about originality.
>
> **A1**. Thank you for your comment. We think our SPARK differs significantly from DGODE[1].
>
> - **Difference with discrete coding.** DGODE is an end-to-end framework. It utilizes discrete "codebank" for **disentangling** environment features and minimizing their impact on node representations. While, SPARK is a upstream-downstream paradigm. In upstream phase, except observations, we incorporate physical parameters and boundary information to train a discrete, physics-rich memory bank. In downstream phase, we forze memory bank's weights and use it to enable targeted augmentation.
>
> - **Difference with graph ODE.** The use of graph ODEs is motivated by their ability to handle irregular data and efficiently perform multi-step temporal extrapolation. However, SPARK and DGODE differ significantly in their details: (1) DGODE uses an RNN architecture to compress historical states, while we use an attention mechanism to compress them into an initial state; (2) we incorporate Fourier blocks into the Graph ODE to enhance the capture of global spectral features, improving spatiotemporal generalization.
>
> To further address your concerns, we run DGODE's open-source code and conduct comparative experiments in both non-OOD (ID) and OOD scenarios. The results shown below indicate that SPARK performs better. We will include these in our revised version.
>
> | Dataset | Prometheus (ID) | Prometheus (OOD) | ERA5 (ID) | ERA5 (OOD) | Spherical-SWE (ID) | Spherical-SWE (OOD) |
> |-------------|---------|--------|---------|---------|---------|---------|
> | DGODE       | 0.0344  | 0.0359 | **0.0387**  | 0.0435  | 0.0024  | 0.0029  |
> | Ours        | **0.0323**  | **0.0328** | 0.0398 | **0.0401**  | **0.0022** | **0.0024** |
>
> ---
> > **Q2**. The algorithm is not clearly presented.
>
> **A2**. Thank you for your detailed comments. We have taken steps to address these issues in the revised manuscript.
>
> - **About boundary representation.** In our paper, the "real boundary" ($\boldsymbol{p}^{boun}_i$) refers to the relative positional relationship between node $i$ and the nearest boundary point. We combine this with the node $i$'s coordinate information $\mathbf{s}_i$ to obtain the position encoding $\boldsymbol{p}^{rel}_i$. We have made detailed corrections to Equation (1), as shown below. For a more intuitive understanding, we use the ERA5 dataset as an example and visualize the boundary information in Appendix I.
>
> $\boldsymbol{u}_i = \text{Proj} \left( \mathcal{X}_i , \boldsymbol{p}^{rel}_i \right) \quad \text{with} \quad
>     \boldsymbol{p}^{rel}_i = \phi\left( \mathbf{s}_i, \boldsymbol{p}^{boun}_i \right).$
>
> - **Time index.** We consistently follow the notation in the Problem Definition. The length of history observations is $T_0$, and the length of future predictions is $T$. We have revised Equations (8) and (11) accordingly, as shown below.
>
> $\alpha_{i}^{t}= \left(\mathcal{v}\_{i}^{t} \right )^{T} \cdot \mathrm{tanh}\left(\left(\frac{1}{T\_0} \sum\{t=1}^{T\_0} \mathcal{v}\_{i}^{t}\right)W\_{\alpha}\right), \quad (8)$
>
> $\mathcal{L}\_{\text{dyn}} = \frac{1}{TN} \sum\_{i=1}^T \sum\_{i=1}^N \|\hat{\mathcal{Y}}\_{i}^{t} - \mathcal{Y}\_{i}^{t}\|\_2^2 + \lambda\_{\text{reg}} \mathcal{R}(\theta). \quad (11)$
>
> - **Index notation.** We have revised the index notation in the pretraining loss equation (6) to ensure it is accurate and meaningful, which is shown below.
>
> $\mathcal{L}\_{pre}=\frac{1}{T N} \sum\_{t=1}^T \sum\_{i=1}^N\left(\hat{\mathcal{X}}\_{i}^{t}-\mathcal{X}\_{i}^{t}\right)^2+ \frac{1}{T N} \sum\_{t=1}^T \sum\_{i=1}^N\left (\mu \left\|\mathcal{h}\_{i}^{t}-\mathbf{s g}[\mathcal{e}]\right\|\_2^2+\gamma\left\|\mathbf{s g}\left[\mathcal{h}\_{i}^{t}\right]-\mathcal{e}\right\|\_2^2\right ).  \quad (6)$
>
> Further, we have thoroughly reviewed the entire manuscript and standardized the writing to ensure consistency in symbols and formulas.
>
> ---
> [1] Wu H, et al. "Prometheus: Out-of-distribution Fluid Dynamics Modeling with Disentangled Graph ODE." ICML2024.

---

> ### Author Response · Authors · 2024-11-21
> **Response to Reviewer 89BR (Part 2/3)**
>
> > **Q3**. It's unclear how the discrete memory bank is built. What are the e_i in E and how are they constructed?
>
> **A3**. Thank you for your feedback. SPARK builds its discrete memory bank using a training strategy inspired by VQ-VAE[1]. Each $e_i$ in the memory bank $E =${$ \{e_1, e_2, \dots, e_M\} $}$ \in \mathbb{R}^{M \times D}$ represents an embedding vector, where $M$ is the fixed size of the memory bank, manually set as a hyperparameter. These embeddings are initialized randomly at the start of training.
>
> > **Q4**. How are the with and without OOD datasets constructed in the experiments? Is there an explanation for why SPARK achieved better performance on OOD cases even than other models did on non-OOD cases?
>
> **A4**. Thank you for your comment.
> - **Experimental settings.** For dataset setting, we propose that training and testing in the in-domain parameters is called w/o OOD experiments, while training in the in-domain parameters and testing in the out-domain parameters is called w/ OOD experiments. Here we present the in-domain and out-domain parameters for different benchmarks in the table below.
>
> | Benchmarks | In-Domain Parameters   | Out-Domain Parameters   |
> |------------|------------------------|-------------------------|
> | PROMETHEUS | $(a_1, a_2, \ldots, a_{25})$, $(b_1, b_2, \ldots, b_{20})$ | $(a_{26}, a_{27}, \ldots, a_{30})$, $(b_{21}, b_{22}, \ldots, b_{25}\)$ |
> | 2D Navier-Stokes Equation       | $ν = (1e^{-1}, 1e^{-2}, \ldots, 1e^{-7}, 1e^{-8}) $ | $ν = (1e^{-9}, 1e^{-10})$ |
> | Spherical Shallow Water Equation  | $ν = (1e^{-1}, 1e^{-2}, \ldots, 1e^{-7}, 1e^{-8}) $   | $ν = (1e^{-9}, 1e^{-10}) $   |
> | 3D Reaction-Diffusion Equations   | $D = (2.1 × 10^{-5}, 1.6 × 10^{-5}, 6.1 × 10^{-5})$ | $D = (2.03 × 10^{-9}, 1.96 × 10^{-9}) $ |
> | ERA5  | $V = ({Sp, SST, SSH, T2m})$  | $V = ({SSR, SSS})$ |
>
>
> - **Explanation for SPARK's better performance.** Our SPARK is specifically designed for OOD problem. SPARK's physics-guided enhancement improves generalization by leveraging physical priors. This reduces sensitivity to distribution shifts. Additionally, the fourier-enhanced graph ODE module provides a robust mechanism for prediction, outperforming other baselines. To address your concern, we use three models specialized in OOD dynamical system modeling (LEADS[2], CODA[3], NUWA[4]), along with FNO, for comparison. The results shown below indicate that OOD-specific models outperform FNO in both OOD and non-OOD scenarios, with SPARK achieving the best performance. We will include these in our revised version.
>
> | Dataset     | Prometheus (ID) | Prometheus (OOD) | ERA5 (ID) | ERA5 (OOD) | Spherical-SWE (ID) | Spherical-SWE (OOD) |
> |-------------|----------------|----------------|---------|----------|---------|----------|
> | FNO         | 0.0547         | 0.0606         | 0.7233  | 0.9821   | 0.0061  | 0.0084   |
> | LEADS       | 0.0374         | 0.0403         | 0.2367  | 0.4233   | 0.0038  | 0.0047   |
> | CODA        | 0.0353         | 0.0372         | 0.1233  | 0.2367   | 0.0034  | 0.0043   |
> | NUWA        | 0.0359         | 0.0398         | 0.0645  | 0.0987   | 0.0032  | 0.0039   |
> | Ours        | **0.0323**  | **0.0328** | **0.0398** | **0.0401**  | **0.0022** | **0.0024** |
>
> ---
> [1] Van Den Oord A, et al. "Neural discrete representation learning." NeurIPS2017.
>
> [2] Kirchmeyer M, et al. "Generalizing to new physical systems via context-informed dynamics model." ICML 2022.
>
> [3] Yin Y, et al. "LEADS: Learning dynamical systems that generalize across environments." NeurIPS2021.
>
> [4] Wang K, et al. "NuwaDynamics: Discovering and Updating in Causal Spatio-Temporal Modeling." ICLR2024.

---

> > ### Author Response · Authors · 2024-11-21
> > **Response to Reviewer 89BR (Part 3/3)**
> >
> > > **Q5**. Given the big accuracy difference, what is the training cost comparison?
> >
> > **A5**. Thank you for your comment. We add experiments of computational costs below. To be fair, we conduct the experiments on a single NVIDIA 40GB A100 GPU. From the results, we can observe that our method has a competitive computation cost. We will include it in our revised version.
> > | Method       | UNet | ResNet | VIT   | SwinT | FNO  | UNO  | CNO  | NMO  | Ours |
> > |--------------|------|--------|-------|-------|------|------|------|------|------|
> > | Training time (h) | 11.2  | 9.76   | 14.5  | 12.3   | 6.9  | 7.8  | 13.4 | 6.3  | 6.7  |
> > | Inference time (s) | 1.34  | 0.93   | 1.32  | 1.13   | 0.54 | 0.67 | 0.12 | 0.52 | 0.55  |
> >
> > > **Q6**. What are the training details, e.g., model architecture, optimizer, training devices? Is boundary information injected at two places, i.e., through node features and directly through the boundary latent vector B?
> >
> > **Q6**. Thank you for your comment. Training details are as follows:
> > - **Model architecture.** To help you better understand our model architecture, we have provided detailed structural information using the Navier-Stokes equations as an example, as shown in the table below.
> >
> > | Upstream | | | Downstream |  |  |
> > | - | - | - | - | - | - |
> > | Procedure | Layer | Dimention | Procedure | Layer | Dimention |
> > |Boundary information injection| Boundary Fusion (Concat + Linear) | (4096, 128) | Augmentation | GNN Encoder | ($T_0$, 4096, 128) |
> > | |Boundary Encoding (Linear + LayerNorm) | (4096, 128) |  | Memory bank retrival | ($T_0$, 4096, 128) |
> > | Physical parameters injection|Channel attention | (2, 128) | Historical observations encoding | Attention score of time steps | (, $T_0$) |
> > |       |Aggregation | (4096, 128) | | Initial state encoding | (1, 4096, 128)
> > |GNN  reconstruction   |Graph Encoder (GNN Layer × L)| (4096, 128) | Fourier-enhanced graph ODE | Fourier transform | (1, 4096, 128) |
> > |  | BatchNorm + ReLU | (4096, 128) | | Linear transform | (1, 4096, 128) |
> > | Memroy bank | Construction | (M, 128) | | Inverse Fourier transform | (1, 4096, 128) |
> > |  | Linear + LayerNorm | (4096, 128) | | ODE solver | (T, 4096, 128) |
> >
> > - **Optimizer and training devices.** We use the **Adam** optimizer for training. Training is conducted on **8 NVIDIA 40GB A100 GPUs**, and inference is performed on **a single NVIDIA 40GB A100 GPU**.
> >
> > - **About boundary information.** We acknowledge that boundary information is injected at two places. On one hand, we integrate boundary location information $\boldsymbol{p}^{boun}_i$ with node features. On the other hand, we inject the latent vector $\mathcal{B}$ encoded from boundary information into the message-passing layers of the GNN. Details are in Equantion (1) and (4). To facilitate understanding, we provide a schematic of the boundary information in Appendix I.
> >
> > ---
> > Thanks again for your valuable feedback! Please let us know if you have further questions.
> >
> > Best,
> >
> > the Authors

---

> > > ### Author Response · Authors · 2024-11-25
> > > **Thank you & Looking forward to further discussion**
> > >
> > > Dear Reviewer 89BR,
> > >
> > > We would like to extend heartfelt thanks to you for your time and efforts in the engagement of author-reviewer discussion. To facilitate better understanding of our rebuttal and revision, we hereby summarize your key concerns and our responses as follows:
> > >
> > > - **About the novelty of SPARK.**
> > > We have clearly explained the differences from the paper you mentioned and provided corresponding comparison experiments.
> > >
> > > - **About the presentation of algorithm and symbols.**
> > > We have thoroughly reviewed the entire manuscript and standardized the writing to ensure consistency in symbols and formulas.
> > >
> > > - **About the construction of the discrete memory bank.**
> > > We have provided details about construction of the discrete memory bank and the embeddings within it.
> > >
> > > - **About the details of OOD experimental setup, training cost, and training details.**
> > > We have included details and additional experiments about what you mentioned. Also, we have added relevant contents in the revised appendix.
> > >
> > > For other issues not mentioned here, please refer to our detalied rebuttal response. We sincerely hope this addresses your concerns! We respectfully look forward to further discussion with you.
> > >
> > > Thank you again for your valuable guidance and thoughtful review.
> > >
> > > Warm regards,
> > >
> > > Authors

---

> > > > ### Author Response · Authors · 2024-11-28
> > > > **Thank you & Looking forward to further discussion!**
> > > >
> > > > Dear Reviewer 89BR,
> > > >
> > > > We sincerely thank you for your valuable and constructive feedback! Since the Discussion Period Extension provides us with additional time, we are eager to address any further concerns you may have. If our current response satisfactorily resolves your main concerns, we kindly ask for your reconsideration of the score. Should you have any further advice on the revised paper and/or our rebuttal, please let us know, and we will be more than happy to engage in further discussion and improve the paper.
> > > >
> > > > Thank you so much for devoting time to improving our paper!
> > > >
> > > > Best regards,
> > > >
> > > > the Authors

---

> > > > > ### Comment · Reviewer_89BR · 2024-12-02
> > > > >
> > > > > Thank you for addressing my comments. While I still have doubts about the method's novelty and suggest polishing the paper with clear variable definitions and method descriptions, I appreciate the extensive experiments conducted by the authors and will raise my score to 6.

---

> ### Author Response · Authors · 2024-12-02
> **Thank You for Your Feedback and Support**
>
> Dear Reviewer 89BR,
>
> Thank you for your valuable feedback and we will polish the paper based on your suggestions. We greatly appreciate your recognition of our extensive experiments and your support in raising the score!
>
> Warm regards,
>
> Authors

---

### Official Review · Reviewer_Rucb · 2024-11-04

**Soundness:** 4
**Presentation:** 4
**Contribution:** 4
**Rating:** 8
**Confidence:** 3

**Summary:**

The paper proposes a new approach to spatiotemporal surrogate modeling. Their approach aims to target some of the limitations of data-driven models as they pertain to distribution shift. The framework, SPARK, combines physics-guided data augmentation and compression to enhance generalization. Key architectural innovations include a discrete memory bank for storing previous physical samples, physical prior and BC incorporation with graph neural nets, and a curriculum learning strategy to incorporate augmented data progressively. SPARK's superior performance is then evaluated on a relatively large suite of benchmark datasets.

**Strengths:**

- The incorporation of a data bank to progressively generate augmented samples is an intelligent design choice, enabling the model to store physical information for use in OOD prediction. Combined with the work on storing physical parameters and boundary conditions, the authors have presented many useful tricks for physical surrogate modeling.
- Extensive tests across numerous datasets and benchmarks conclusively demonstrate the superior performance of this training strategy, particularly in OOD scenarios. The benchmarks are extensive as well, and helpful in framing the work.
- The theoretical framework is helpful, providing solid support for the model's architecture and approach.
- The paper is well-written and well-presented.

**Weaknesses:**

- Many new architectural design choices are proposed (handling of physical parameters, boundary conditions, data banks, curriculum learning, etc.). However, it is unclear how much each strategy contributes to the success of the model, and some ablation studies would be useful.

**Questions:**

- Can the data bank be used for direct retrieval-augmentation?
- How well does the model perform in the very low-data regime (just a few samples for transfer learning)?
- Have the authors explore generalization to 1-D or 3-D data at all?

---

> ### Author Response · Authors · 2024-11-21
> **Response to Reviewer Rucb (Part 1/2)**
>
> Dear Reviewer Rucb,
>
> We sincerely appreciate the time you’ve dedicated to reviewing our paper, as well as your valuable insights and support. Your positive feedback is highly motivating for us. Below, we address your primary concern and offer further clarification.
>
> ---
> > **Q1**. It is unclear how much each strategy contributes to the success of the model, and some ablation studies would be useful.
>
> **A1**. Thanks for your valuable feedback. To further demonstrate the contribution of each strategy, we conduct ablation experiments with five model variants. The experiments are conducted on Prometheus and Navier–Stokes datasets with OOD scenarios, and the results are shown below.
>
> | | Ours | w/o parameter | w/o boundary | w/o parameter\&boundary | w/o memory bank | w/o curriculum learning |
> | ------------- | ---------- | ------ | ------ | ------ | ------ | -- |
> | Prometheus    | **0.0301** | 0.0357 | 0.0324 | 0.0397 | 0.0416 | 0.0338 |
> | Navier–Stokes | **0.0725** | 0.0833 | 0.0764 | 0.0902 | 0.1058 | 0.0792 |
>
> As observed, removing physical parameters or boundary conditions during pretraining leads to a performance decline, with an even greater drop when the memory bank is not used. This validates the effectiveness of physical compression when addressing OOD problems. We will include these in our revised manuscript.
>
> ---
> > **Q2**. Can the data bank be used for direct retrieval-augmentation?
>
> **A2**. Thank you for your comment. We acknowledge that the memory bank in our SPARK framework can be used for direct retrieval augmentation. After pre-training, the memory bank’s parameters are frozen, allowing new samples to retrieve physics-rich embeddings in the bank for augmentation. For convenient understanding, we have illustrated this with diagram in Appendix H.9.

---

> > ### Author Response · Authors · 2024-11-21
> > **Response to Reviewer Rucb (Part 2/2)**
> >
> > > **Q3**. How well does the model perform in the very low-data regime (just a few samples for transfer learning)?
> >
> > **A3**. Thanks for your feedback. We conduct experiments on the performance of our model in very low-data regime. Specifically, after pre-training on the full ERA5 dataset, we finetune on subsets of the Sevir dataset with varying amounts of data (1\%, 3\%, 5\%, and 10\%). Below is a detailed comparison of baseline models (PredRNN and SimVP) with and without the SPARK plugin.
> >
> > |               | 1\% Sevir | 3\% Sevir | 5\% Sevir | 10\% Sevir |
> > |---------------|------------|-----------|-----------|------------|
> > | PredRNN       | 3.51→3.38  | 2.57→2.35 | 1.83→1.68 | 1.22→1.16  |
> > | PredRNN+SPARK | 3.37→3.02  | 2.49→2.14 | 1.72→1.45 | 1.14→0.97  |
> > | Simvp         | 2.43→2.20  | 1.86→1.55 | 1.29→1.11 | 0.75→0.68  |
> > | Simvp+SPARK   | 2.30→1.98  | 1.75→1.23 | 1.21→0.98 | 0.71→0.57  |
> >
> > The results show that models with SPARK plugin consistently outperform their baseline models even in very low data regime.
> >
> > In addition, we conduct **zero-shot** experiments on the Navier-Stokes (NS) Equations. Following the setup of Li et al.[1], we train on 64×64 NS Equations with Reynolds number of 1e-4 and directly tested on 128×128 NS Equations. The results below demonstrate that our SPARK plugin possesses well transfer learning capability.
> >
> > |            | Zero-shot   |
> > | ---------- | ----------- |
> > | FNO        | 0.274→0.251 |
> > | FNO+SPARK  | 0.256→0.223 |
> >
> > ---
> > > **Q4**. Have the authors explore generalization to 1-D or 3-D data at all?
> >
> > **A4**. Thank you for your comment. We have conducted experiments on 3D Reaction-Diffusion Equations in our paper. Here, we add experiments on 1-D data using the Burgers Equations[2]. The results are shown below, which indicate that our method is also applicable to 1-D data.
> >
> > |            | U-Net | ResNet |  FNO  |  CNO  |  NMO  | Ours+SPARK |
> > | ---------- | ----- | ------ | ----- | ----- | ----- | ---------- |
> > |  w/o OOD   | 0.362 |  0.338 | 0.298 | 0.314 | 0.246 | **0.228**      |
> > |  w/ OOD    | 0.397 |  0.351 | 0.325 | 0.338 | 0.273 | **0.243**      |
> >
> > Additionally, we select FNO, CNO, and NMO as baselines to evaluate SPARK's generalization capability across different dimensional data. Specifically, we pre-train on 2-D Navier-Stokes Equations and finetune on 1-D Burgers Equations. The results below validate that model variants with SPARK plugin have better generalization capability than their baseline models. We will include it in our revised version.
> >
> > |          | FNO         | FNO+SPARK   | CNO         | CNO+SPARK   | NMO         | NMO+SPARK   |
> > | -------- | ----------- | ----------- | ----------- | ----------- | ----------- | ----------- |
> > | Burgers  | 0.317→0.294 | 0.308→0.275 | 0.298→0.275 | 0.280→0.256 | 0.241→0.223 | 0.228→0.204 |
> >
> > ---
> > [1] Li, Z, et al. "Fourier neural operator for parametric partial differential equations." ICLR2021.
> >
> > [2] Takamoto M, et al. "Pdebench: An extensive benchmark for scientific machine learning." NeurIPS2022.
> >
> > ---
> > Thanks again for appreciating our work and for your constructive suggestions! Please let us know if you have further questions.
> >
> > Best,
> >
> > the Authors

---

> > > ### Comment · Reviewer_Rucb · 2024-11-24
> > >
> > > The authors have done a good job of addressing my concerns. I am particularly happy to see the improvement in performance in the low data and zero-shot regime. I will keep my score at an 8 and recommend this paper be accepted.

---

> > > > ### Author Response · Authors · 2024-11-24
> > > > **Thanks for your recognition!**
> > > >
> > > > Dear Reviewer Rucb,
> > > >
> > > > We sincerely appreciate your valuable feedback and recognition! We are pleased to know that your concerns have been addressed! We will definitely incorporate your suggestions into our revised version. Please kindly let us know if you have any questions further!
> > > >
> > > > Best regards,
> > > >
> > > > the Authors

---

### Author Response · Authors · 2024-11-21
**General Response**

Dear Reviewers,

Thanks for your time and valuable feedbacks. We acknowledge three reviewers (Reviewer Rucb, Reviewer 89BR, and Reviewer pH3K) comment that the work is novel or nice. We acknowledge the positive comments such as "a new approach" (Reviewer Rucb), "superior performance" (Reviewer Rucb), "an intelligent design choice" (Reviewer Rucb), "many useful tricks" (Reviewer Rucb), "benchmarks are extensive" (Reviewer Rucb), "the theoretical framework is helpful" (Reviewer Rucb), "well-written and well-presented", "the idea is nice" (Reviewer 89BR), "a good number of benchmarks" (Reviewer 89BR), "crucial for real-world applications" (Reviewer bjTD), "extensive experimental results" (Reviewer bjTD), "an interesting idea" (Reviewer pH3K), "important topics" (Reviewer pH3K), "well-written and has a detailed presentation" (Reviewer pH3K). We have also responded to your concerns in the following.

Please let us know if you have any additional questions or concerns. We will try our best to address them.

Best regards,

the Authors

---

### Author Response · Authors · 2024-11-23
**Summary of Response**

Dear Reviewers,

Thank you for your thorough and insightful reviews. We sincerely appreciate your feedback, which has significantly enhanced our paper! Below, we summarize the key concerns raised and our corresponding responses:

- **Implementation details of SPARK** (Reviewer 89BR, bjTD, pH3K)

    We have presented our model details in tabular form and used the Navier-Stokes equation as an example to show dimensional changes and related experiments.

- **Problem about novelty** (Reviewer 89BR, bjTD)

    We have clarified the differences between our SPARK and the mentioned models. Furthermore, we have conducted comprehensive experiments to verify this.

- **Contribution of each component or strategy** (Reviewer Rucb, pH3K)

    We have added ablation experiments to confirm the effectiveness of each component or strategy.

- **Details of boundary information and OOD experimental setup** (Reviewer 89BR, bjTD, pH3K)

    We have included a more detailed description of the boundary information, and an intuitive figure is shown in Appendix I. For OOD experimental setup, we have provided a clear table to present.

Once again, we are truly grateful for your valuable feedback and are happy to address any further concerns or questions!

Sincerely,

Authors

---

### Meta-Review · Area_Chair_JiYn · 2024-12-19

**Metareview:**

This paper introduces SPARK, a physics-guided quantized augmentation plugin for dynamical system modeling. While traditional methods are computationally expensive and sensitive to initial conditions, and current deep learning models depend on large datasets and stable distributions, SPARK overcomes these issues. It uses a reconstruction autoencoder to build a physics-rich memory bank and applies attention mechanisms and Fourier-enhanced graph ODEs for efficient long-term predictions. Extensive experiments show that SPARK outperforms baseline methods in handling data scarcity and distribution shifts.

Strengths:

The incorporation of a data bank to progressively generate augmented samples is an insightful design choice.

The paper presents extensive experimental results across a variety of benchmark datasets.

Weaknesses:

The degree of originality and novelty is relatively low when compared to methods like DGODE and BeamVQ.

The motivation behind the approach and the algorithm itself are not clearly articulated.

It is unclear how much each individual strategy contributes to the overall success of the model. Some ablation studies would provide valuable insights here.


While the proposed approach is interesting and the experimental results cover a broad range of benchmarks, the novelty of the methodology is only marginally significant. Long-term prediction is a common metric in dynamical system modeling, yet the experiments in this paper are limited to 10-50 steps. For PDE modeling, it would be beneficial to evaluate rollout errors for more than 100 ane even 1000 steps [1]. Given these concerns, I suggest a borderline reject but encourage the authors to address the reviewers' feedback and resubmit to a top conference.

**Additional Comments On Reviewer Discussion:**

During the rebuttal period, the authors addressed the following points:

Ablations: Reviewers Rucb, bjTD, and pH3K requested additional ablation studies. The authors provided comprehensive experiments and effectively addressed these concerns.

Novelty: Reviewers 89BR and bjTD questioned the novelty of the approach. The authors clarified their methodology with additional explanations and experimental results that differentiate their work from the methods cited by the reviewers. After reviewing both the authors' responses and the paper, I tend to agree that while the overall architecture is novel, the technical originality of individual components appears incremental.

Clarity of Presentation: Reviewers 89BR, bjTD, and pH3K initially had reservations about the clarity of the presentation. The authors provided additional details and improved the overall clarity of the paper.

Plagiarism Concern: Reviewer bjTD raised a potential plagiarism issue, comparing the paper to another ICLR 2025 submission. However, after comparing the tasks, methodologies, and experiments, I believe these are distinct papers.

In summary, the authors have addressed most of the concerns raised by the reviewers. However, reviewer bjTD still holds reservations regarding the novelty. In my opinion, the novelty of the methodology is only marginally significant. More importantly, the significance of the work could be improved by extending the evaluation to long-term predictions, particularly beyond 1000 steps.

[1] Encoding physics to learn reaction-diffusion processes, Nature Machine Intelligence. Nature Machine Intelligence, 5(7):765–779, 2023.

---

### Decision · Program_Chairs · 2025-01-22

Reject